# Okapi: Generalising Better by Making Statistical Matches Match

**Myles Bartlett**[1*]    **Sara Romiti**[1]    **Viktoriia Sharmanska**[1,2]    **Novi Quadrianto**[1,3,4]

[1]Predictive Analytics Lab, University of Sussex    [2]Imperial College London
[3]BCAM Severo Ochoa Strategic Lab on Trustworthy Machine Learning
[4]Monash University, Indonesia

## Abstract

We propose *Okapi*, a simple, efficient, and general method for robust semi-supervised learning based on online statistical matching. Our method uses a nearest-neighbours-based matching procedure to generate cross-domain views for a consistency loss, while eliminating statistical outliers. In order to perform the online matching in a runtime- and memory-efficient way, we draw upon the self-supervised literature and combine a memory bank with a slow-moving momentum encoder. The consistency loss is applied within the feature space, rather than on the predictive distribution, making the method agnostic to both the *modality* and the *task* in question. We experiment on the WILDS 2.0 datasets [63], which significantly expands the range of modalities, applications, and shifts available for studying and benchmarking real-world unsupervised adaptation. Contrary to [63], we show that it is in fact possible to leverage additional unlabelled data to improve upon empirical risk minimisation (ERM) results with the right method. Our method outperforms the baseline methods in terms of out-of-distribution (OOD) generalisation on the iWildCam (a multi-class classification task) and PovertyMap (a regression task) image datasets as well as the CivilComments (a binary classification task) text dataset. Furthermore, from a qualitative perspective, we show the matches obtained from the learned encoder are strongly semantically related. Code for our paper is publicly available at `https://github.com/wearepal/okapi/`.

## 1 Introduction

Machine learning models have been deployed for safety-critical applications such as disease diagnosis [73] and self-driving cars [77], and in socially important contexts such as the allocation of healthcare, education, and credit (e.g. [23, 35]). Many machine learning algorithms, however, rely on supervision from a large amount of labelled data, and are typically trained to exploit complex relationships and distant correlations present in the training dataset. This strategy has proven to be effective in the setting when we have training (source) and test (target) data that are i.i.d.

In reality, machine learning models are often deployed on target data whose distribution is different from the source distribution they were trained on. For example, in the task of classifying animal species in a camera trap image, one aims to learn a model that can generalise to new camera trap locations despite variations in illumination, background, and label frequencies, given training examples from a limited set of camera trap locations. Exploiting correlations that only hold in these limited locations but not in the new locations can hurt out-of-distribution (OOD) generalisation. While we only have a small subset of camera traps that have their images labelled, we have a large

---

*Corresponding author: m.bartlett@sussex.ac.uk.

amount of unlabelled data from the other camera traps that capture diverse operating conditions. In general, unlabelled data is much more readily available than labelled data and can often be obtained from distributions beyond the source distribution. Taking advantage of these unlabelled data during training is a key element to build robust models that have good OOD performance without sacrificing in-distribution (ID) performance.

Our work is a direct response to the empirical conclusion of [63] for the WILDS 2.0 dataset that existing semi-supervised methods – leveraging the unlabelled data provided in this extended version of the WILDS benchmark [41] – fail to provide consistent benefit over the combination of judicious data-augmentation and standard empirical risk minimisation (ERM). We show that with the right method, however, it is in fact possible to make effective use of large volumes of unlabelled data as supplement to a smaller set of labelled data, from a limited set of domains, to achieve strong generalisation to data from domains outside the training distribution. To develop this method, we turn to a statistical matching (SM) framework [57, 59, 60], a model-based approach for providing joint information on variables and indicators collected through multiple sources. SM has been widely utilised to assess the effect of interventions in numerous fields, such as education, medical and community policies (e.g. [10, 17]). In SM, intervened units are paired with control units and those units without a sufficiently-good match according to a given statistical criteria are excluded when estimating the treatment effect. In the running example of animal-species classification, intervened units may correspond to the limited set of camera trap locations that are fully-annotated, while control units refer to the many more camera trap locations that are only partially annotated. Pairing is beneficial for capturing diverse operating conditions, yet the ability to drop unpaired units is crucial for mitigating the risk of statistically-poor matches corrupting the training signal.

In designing an online method for statistically matching samples from different domains (camera-trap locations) and using this to define a consistency loss, we arrive at our proposed semi-supervised method, *Okapi*. This consistency loss is predicated on the simple idea of pulling together similar samples from different domains within the latent space of the encoder, and using this to bootstrap said encoder such that the distributions become progressively more aligned over the course of training. Since matching samples using the full dataset at each step of training is computationally infeasible, we instead approximate it using a combination of momentum-encoding and a memory-bank that has been well-proven in self-supervised learning [33, 42]. Compared with other consistency-based methods such as FixMatch [67], Okapi has the advantage of being agnostic to both the task and the modality, in addition to being distributionally robust. Contrary, to Sagawa et al. [63], we show that the supplementary unlabelled data and domain information can be leveraged by Okapi to improve upon standard ERM on datasets from the WILDS 2.0 benchmark.

## 2  Preliminaries

### 2.1  Problem setting

In the standard supervised setting, one is given a dataset, $\mathcal{D}_l \triangleq \{x_i, y_i\}_{i=1}^{N_l}$, and trains a model, parameterised by $\theta$, to well-approximate the empirical distribution as $p_\theta(y|x)$. Labelled data is limited by the cost of annotation yet one often has access to a far larger corpus of unlabelled data, $\mathcal{D}_u \triangleq \{x_i\}_{i=1}^{N_u}$, which can be used to supplement $\mathcal{D}_l$. Semi-supervised learning is motivated by the idea that this additional data can often be used to improve the ID and/or OOD performance of $p_\theta(y|x)$. We can view unsupervised domain adaptation (UDA) as a special case of semi-supervised learning, where there is assumed to be some distribution shift (adverse to a naïvely-trained predictor) between $\mathcal{D}_l$ and $\mathcal{D}_u$. Here, $\mathcal{D}_u$ comes from the domain on which $p_\theta(y|x)$ is to be evaluated, such that we have $\mathcal{D}_u \triangleq \mathcal{D}_{\mathrm{OOD}}$, where $\mathcal{D}_{\mathrm{OOD}}$ denotes the target domain, that is OOD w.r.t. $D_l$. In the most general sense, a *domain*, or *environment* [4, 19] describes some partitioning of the data according to its source or some secondary characteristic, such as time of day, weather, location, lighting, or the model of the device used to collect said data; one would hope that a predictor trained under one set of conditions (e.g. day) would perform with minimal degradation under another set of conditions (e.g. night) when those conditions are irrelevant to the task at hand.

Assuming the data follows the conditional generative distribution $x \sim p(x|s)$, where $s$ is the domain label, one would ideally use $\mathcal{D}_{\mathrm{OOD}}$ to learn invariance to the marginal distribution, $p(s)$, and thereby achieve the equivalence $p_\theta(y|x) = p_\theta(y|x, s)$. In practice, one typically does not have access to $\mathcal{D}_{\mathrm{OOD}}$ but does have access to training data sourced from a mixture of domains which can be

leveraged to learn a more general invariance that extends to those domains outside the training distribution [4]. Such a learning paradigm is referred to as domain generalisation (DG). While some DG works consider the more extreme case of $s$ being unobserved [19], we follow the more conventional setup [4, 43, 63] in which the domain(s) associated with each sample (labelled and unlabelled) is indicated by the discrete label (set of labels) $s$. We denote the set of possible domains for the in-distribution labelled and unlabeled data, as $\mathcal{S}_l$ and $\mathcal{S}_u$, respectively, and their union as $\mathcal{S} \triangleq S_l \cup S_u$. Following the setup established in [63], $\mathcal{D}_u$ is assumed to be unlabelled only w.r.t the targets and not w.r.t the domain labels and thus that both $\mathcal{D}_l$ and $\mathcal{D}_u$ can be augmented with the latter to give the re-definitions $\mathcal{D}_l \triangleq \{x_i, y_i, s_i\}_{i=1}^{N_l}$ and $\mathcal{D}_u \triangleq \{x_i, s_i\}_{i=1}^{N_u}$.

## 2.2 Statistical matching

Statistical matching is a sampling strategy which aims to balance the distribution of the observed covariates in the *treated* and *control* groups. In general terms, observed covariates $x$ are measured characteristics of the samples; in our work we refer to the encodings generated by a deep neural network as covariates instead of the original characteristics. The treated and control groups are two partitions of the data; specifically, the treated group is the set of samples having a specific value of a variable of interest (here, the domain indicator, $s$) and the control group is its complement.

In this work we utilise Nearest Neighbour (NN) matching, a distance-based matching method that pairs sample $i$ of the treated group with the closest sample $j$ belonging to the control group. A distance measure is used to define how close two samples, $i$ and $j$, are, with *propensity score distance* (PSD) and *Euclidean distance* being two widely-used distances that we employ here – indirectly (as a means of filtering) and directly, respectively.

The propensity score distance is defined as the difference between propensity scores, $e_i$ and $e_j$, of samples $i$ and $j$, i.e. $\mathrm{PSD}(e_i, e_j) \triangleq |e_i - e_j|$. In causal inference, the propensity score refers to the probability of sample $i$ belonging to the treated group, given its covariates $x_i$ [58]; in practice, this conditional probability is rarely known a priori and thus requires estimation, typically via logistic regression [68]. We generalise the notion of a propensity score to categorical domains simply by modelling the conditional probability for each domain, with $e_i$ instead a $|\mathcal{S}|$-dimensional probability vector. The Euclidean-distance approach, in contrast, computes the distance between the covariates, $x_i$ and $x_j$, of a given pair of samples. Despite PSD being the more prevalent of the two distances, it is ill-suited to cases in where pairs are close in value w.r.t. all covariates and in such cases Euclidean distance should be preferred [40]. Nevertheless, propensity scores remain a relevant component of NN-based matching for defining *calipers* that can reduce the likelihood of false-positive matches.

We make use of two types of caliper, *fixed* and *standard deviation*. The fixed caliper [20], $t_f$, defines a region of common support between the estimated propensity score distribution of the two groups; only those samples within the feasible region are admissible for matching. For binary problems, the feasible region is symmetric such that we have $\{i \mid e_i \in (1 - t_f, t_f)\})$ whereas in the more general, categorical case the constraint is one-sided, i.e. $\{i \mid \|e_i\|_\infty < t_f)\}$. This selection rule helps by removing samples with extreme propensity scores. The standard deviation-based caliper (std-caliper), on the other hand, [59] defines the maximum discrepancy permitted between paired two samples. The discrepancy is usually expressed in terms of estimated PSD as $|e_i - e_j| < \sigma \cdot t_\sigma$, where $\sigma$ denotes the mean of the group-wise standard deviations of the propensity scores and $t_\sigma$ controls the percentage bias-reduction of the covariates. In the categorical case, we can simply substitute the absolute value for the infinity norm: $\|e_i - e_j\|_\infty < \sigma \cdot t_\sigma$. In the following section, we describe how one can leverage this matching framework to define a consistency loss encouraging inter-domain robustness.

## 3 Method

Here, we introduce *Okapi*, a simple, efficient, and general (in the sense that it is applicable to any task *or* modality) method for robust semi-supervised learning based on online statistical matching. Our method belongs to the broad family of consistency-based methods, characterised by methods such as FixMatch [67], where the idea is to enforce similarity between a model's outputs for two views of an unlabelled image. These semi-supervised approaches based on minimising the discrepancy between two views of a given sample are closely related with self-supervised methods based on instance discrimination [15] and self-distillation [6, 14, 30]. Many of the methods within this family, however,

are limited in applicability due to their dependence on modality-specific transformations and only recently has research into self-supervision sought to redress this problem with modality-agnostic alternatives such as MixUp [72], masking [6], and nearest-neighbours [24, 42, 71]. Approaches such as FixMatch, AlphaMatch [28] and CSSL [46] that enforce consistency between the *predictive* distributions suffer further from not being directly generalisable to tasks other than classification. Okapi addresses both of these aforementioned issues through 1) the use of a statistical matching procedure – that we call CaliperNN and detail in Sec. 3.2 – to generate multiple views for a given sample; 2) enforcing consistency between encodings rather than between predictive distributions.

We show that models trained to maximise the similarity between the encoding of a given sample and those of its CaliperNN-generated match are significantly more robust to real-world distribution shifts than the baseline methods, while having the advantage of being both computationally efficient and agnostic to the modality and task in question. Qualitatively speaking, we see that matches produced with the final model are related in semantically-meaningful ways. Furthermore, since the only constraint is that samples be from different domains, the method is applicable whether information about the domain is coarse or fine-grained.

In the following subsections, we begin by giving a general formulation of our proposed semi-supervised loss employing a generic cross-domain $k$-NN algorithm. We then explain how we can replace this algorithm with CaliperNN in order to mitigate the risk of poorly-matched samples, and how the loss may be computed in an online fashion to give our complete algorithm.

## 3.1   Enforcing consistency between cross-domain pairs

We view our predictor as being composed of an encoder (or *backbone*) network, $f_\theta : \mathcal{X} \to \mathbb{R}^d$, generating intermediary outputs (features) $z \triangleq f_\theta(x)$, and a prediction head, $g_\phi$, such that the prediction for sample $x$ is given by $\hat{y} \triangleq g_\phi \circ f_\theta(x)$. We similarly consider the aggregate loss $\mathcal{L}$ as having a two-part decomposition given by

$$\mathcal{L} \triangleq \mathcal{L}_{\text{sup}} + \lambda \mathcal{L}_{\text{unsup}}, \tag{1}$$

where $\mathcal{L}_{\text{sup}}$ is the supervised component measuring the discrepancy (as computed, for example, by the MSE loss) between $\hat{y}$ and the ground-truth label $y$, $\mathcal{L}_{\text{unsup}}$ is the unsupervised component based on some kind of pretext task, such as cross-view consistency, and $\lambda$ is a positive pre-factor determining the trade-off between the two components. For our method, we do not assume any particular form for $\mathcal{L}_{\text{sup}}$ and focus solely on $\mathcal{L}_{\text{unsup}}$.

Given a pair of datasets $\mathcal{D}_l$ and $\mathcal{D}_u$, sourced from the labelled domain $\mathcal{S}_l$, and unlabelled domain $\mathcal{S}_u$ respectively, along with their union $\mathcal{D} \triangleq \mathcal{D}_l \cup \mathcal{D}_u$ our goal is to train a predictor that is robust (invariant) to changes in domain, including those unseen during training. To do this, we propose to regularise $z \triangleq f_\theta(x)$ to be smooth (consistent) within local, cross-domain neighbourhoods. At a high-level, for any given *query* sample $x_q$ sourced from domain $s_q$, we compute $\mathcal{L}_{\text{unsup}}$ as the mean distance between its encoding $z_q \triangleq f_\theta(x_q)$ and that of its $k$-nearest neighbours, $V_k(z_q)$ with the constraint that $\{s_q\} \cap \mathbf{s}_n = \emptyset$, where $\mathbf{s}_n$ is the set of domain-labels associated with $V_k(z_q)$. The general form of this loss for a given sample can then be written as

$$V_k(z_q) \triangleq \text{NN}(z_q, \{f_\theta(x) \mid (x, s) \in \mathcal{D}, s \neq s_q\}, k), \tag{2}$$

$$\mathcal{L}_{\text{unsup}} \triangleq \frac{1}{k} \sum_{z_n \in V_k(z_q)} d(z_q, z_n) \tag{3}$$

where $d : \mathbb{R}^d \times \mathbb{R}^d \to \mathbb{R}$ is some distance function. Here, we follow [30] and define $d$ to be the squared Euclidean distance between normalised encodings for our experiments. Allowing the NN algorithm to select pairs in an unconstrained manner, given the pool of queries and keys, however, can lead to poorly-matched pairs that are detrimental to the optimisation process. To address this, we replace the standard NN algorithm with a propensity-score-based variant, inspired by the statistical matching framework [58].

## 3.2   Cross-domain matching

For the matching component of our algorithm, we propose to use a variant of $k$-NN which, in addition to incorporating the above cross-domain constraint, filters the queries and keys that represent probable

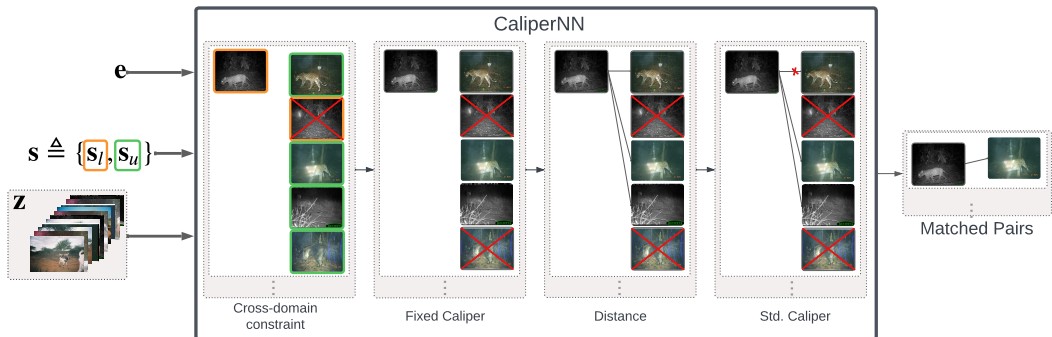

Figure 1: Illustration of our proposed statistical matching algorithm, CaliperNN. Given the anchor image encoding $\mathbf{z}$, the corresponding domain label $s$ (we consider the binary case of labelled vs. unlabelled for simplicity), and propensity score $e$, CaliperNN outputs the closest samples subject to their being from different domains, following filtering by the fixed and std. calipers.

outliers, according to their learned propensity scores. The initial stage of filtering employs a fixed caliper, where samples with propensity scores surpassing a fixed confidence threshold are removed; this is followed by a second stage of filtering wherein any two samples (from different domains) can only be matched if the Euclidean distance between their respective propensity scores is below a pre-defined threshold (std-caliper). See Fig 3.2 for a pictorial representation of these steps and Appendix G for reference pseudocode.

The propensity score, $e$, for a given sample $x$ is estimated as $p(s|z)$ using a linear classifier $f_\theta$, $h_\psi : \mathbb{R}^d \rightarrow \triangle^{|\mathcal{S}|}$ where $\triangle^{|\mathcal{S}|}$ is the probability simplex over possible domain labels, $\mathcal{S}$, induced by the softmax function. $h_\psi^d$ is trained via maximum (weighted) likelihood to predict the domain label of a given sample for all samples within the aggregate dataset $\mathcal{D}$, or (typically) a subset of it, encoded by $f_\theta$. Since we apply both calipers to the learned propensity score, the shape of this distribution can have a significant effect on the outcome of matching. Accordingly, we apply temperature-scaling, with scalar $\tau \in \mathbb{R}_\star^+$, to sharpen or flatten the learned propensity-score distribution. We denote the set of associated parameters ($\{\, t_f, t_\sigma, \tau \,\}$, as the threshold for the fixed-caliper, the threshold for the std-caliper, and the temperature, respectively) as $\xi$ and discuss in Appendix D how one can determine suitable values for these in practice.

For convenience we define the set of all encodings, given by $f_\theta$, as $\mathbf{z} \triangleq \{ f_\theta(x) \,|\, x \in \mathcal{D} \}$, the set of all associated propensity scores as $\mathbf{e} \triangleq \{ h_\psi(x) \,|\, z \in \mathbf{z} \}$, and the set of associated domain labels as $\mathbf{s}$. In the offline case, the matches for $\mathcal{D}$ are then computed as

$$\mathrm{MatchedSamples} \triangleq \{ (z, \mathrm{CaliperNN}_\xi(z, \mathbf{z}, \mathbf{e}, s, k)) \,|\, z \in \mathbf{z}, s \in \mathbf{s} \}, \tag{4}$$

with CaliperNN returning the set of $k$-nearest neighbours according to $d$, subject to the aforementioned cross-domain and caliper-based constraints. We allow for the fact that there may be no valid matches for some samples due to these constraints; in such cases we have $\emptyset$ as the second element of their tuples, indicating that $\mathcal{L}_{\mathrm{unsup}}$ should be set to $0$.

### 3.3 Scaling up with Online Learning

Re-encoding the dataset following each update of the feature-extractor, in order to recompute $\mathrm{MatchedSamples}$, is prohibitively expensive, with cost scaling linearly with $N \triangleq N_l + N_u$. Moreover, CaliperNN requires explicit computation of the pairwise distance matrices, which can be prohibitive memory-wise for large values of $N$. We address these problems using a fixed-size memory bank, $\mathcal{M}_z^{N_\mathcal{M}}$ storing only the last $N_\mathcal{M}$ (where $N_\mathcal{M} \ll N$) encodings from a slow-moving momentum encoder [30, 33], $f_{\theta'}$, which we refer to as the *target* encoder, in line with [30], and accordingly refer to $f_\theta$ as the *online* encoder. Unlike [30], however, we make use of neither a projector nor a predictor head (in the case of the target encoder) in order to compute the inputs to the consistency loss and simply use the output of the backbone as is – this is possible in our setting due to $\mathcal{L}_{\mathrm{sup}}$ preventing representational collapse. More specifically, the target encoder's parameters,

$\theta'$, are computed as a moving average of the online encoder's, $\theta$, with decay rate $\zeta \in (0, 1)$, per the recurrence relation

$$\theta'_t = \zeta\theta'_{t-1} + (1 - \zeta)\theta_t, \tag{5}$$

As the associated domain labels are also needed both for matching and to compute the loss for the propensity scorer, we also store the labels associated with $\mathcal{M}_z^{N_\mathcal{M}}$ in a companion memory bank $\mathcal{M}_s^{N_\mathcal{M}}$. We initialise $\mathcal{M}_z^{N_\mathcal{M}}$ and $\mathcal{M}_s^{N_\mathcal{M}}$ to $\emptyset$, resulting in fewer than $N_\mathcal{M}$ samples being used during the initial stages of training when the memory banks are yet to be populated.

Each iteration of training, we sample a batch of size $B$ from $\mathcal{D}$ consisting of inputs $\mathbf{x}$ and $\mathbf{s}$. During the matching phase, the inputs are passed through the *target* encoder to obtain $\mathbf{z}'_q \triangleq \{f_{\theta'}(x)|x \in \mathbf{x}\}$, serving as the queries for CaliperNN. We also experiment with a simpler variant where the *online* encoder is instead used for this query-generation step, such that we instead have $\mathbf{z}'_q \triangleq \{f_\theta(x)|x \in \mathbf{x}\}$, and find this can work equally well if $\zeta$ is sufficiently high. The keys are then formed by combining the current queries with the past queries contained in the memory bank: $\mathbf{z}_k \triangleq \mathbf{z}'_q \cup \mathcal{M}_z^{N_\mathcal{M}}$. The domain labels associated with $\mathbf{z}_k$ are likewise formed by concatenating the domain labels in the current batch with those stored in $\mathcal{M}_s^{N_\mathcal{M}}$: $\mathbf{s}_k \triangleq \mathbf{s}_q \cup \mathcal{M}_s^{N_\mathcal{M}}$. Once the matches for the current samples have been computed, the oldest $B$ samples in $M_z^{N_\mathcal{M}}$ and $M_s^{N_\mathcal{M}}$ are overwritten with $\mathbf{z}_k$ and $\mathbf{s}_k$, respectively. The consistency loss is then enforced between each query $\mathbf{z}_q \triangleq \{f_\theta(x)|x \in \mathbf{x}\}$, according to the differentiable *online* encoder, and each of its matches, $V_k(z'_q) \triangleq \text{CaliperNN}_\xi(z_q, \mathbf{z}_k, h_\psi(\mathbf{z}_k), \mathbf{s}_k)$ providing that $V_k(z'_q) \neq \emptyset$ (that is, under the condition that the estimated propensity score for $z'_q$ does not violate the caliper(s) and there are at least $k$ valid matches whose estimated propensity scores also do not), with the loss simply 0 otherwise. Since $f_{\theta'}$ is frozen, $\mathbf{z}_k$ carries an implicit stop-gradient and gradients are computed only w.r.t. $\theta$. These steps are illustrated pictorially in Fig 2 and as pseudocode in Appendix G.

Similarly, rather than solving for the optimal parameters, $\psi^\star$ for the propensity scorer given the current values of $\mathbf{z}_k$, which is infeasible for the large values of $N_\mathcal{M}$ needed to well-approximate the full dataset, we resort to a biased estimate of $\psi^\star$. Namely, we train $h_\psi$ in an online fashion to minimise the per-batch loss

$$\mathcal{L}_{\text{ps}} = \frac{1}{|\mathbf{z}_k|} \sum_{z \in \mathbf{z}_k, s \in \mathbf{s}_k} w_{\mathbf{s}_k}(s)\mathcal{H}(h_\psi(z), s), \tag{6}$$

where $\mathcal{H}$ is the standard cross-entropy loss between the predictive distribution and the (degenerate) ground-truth distribution, given by the one-hot encoded domain labels, and $w_{\mathbf{s}_k} : \mathcal{S} \to \mathbb{R}_*^+$ is a function assigning to each $s$ an importance weight [66] based on the inverse of its frequency in $\mathbf{s}_k$ to counteract label imbalance. In the special case in which the $\mathcal{D}_l$ and $\mathcal{D}_u$ are known to have disjoint support over $S$ (that is, $\mathcal{S}_l \cap \mathcal{S}_u = \emptyset$), we can substitute their domain labels with 1 and 0, respectively (such that we have $\mathcal{D}_l \triangleq \{x_i, y_i, 1\}_{i=1}^{N_l}$ and $\mathcal{D}_u \triangleq \{x_i, 0\}_{i=1}^{N_u}$), thus reducing the propensity scorer and CaliperNN to their binary forms. Knowing whether this condition is satisfied a priori (and thus whether the use of domain labels can be forgone completely from our pipeline) is not unrealistic: one may, for example know that two sets of satellite imagery cover two different parts of the world (e.g. Africa and Asia) yet not know the exact coordinates underlying their respective coverage.

## 4  Related Work

**Domain Generalisation**   The goal of domain generalisation (DG) is to produce models that are robust to a wide range of distribution shifts (including those outside the training distribution), given a training set consisting of samples sourced from multiple domains. Despite the various techniques (many well theoretically-motivated) designed to improve the generalisation of deep neural networks current methods continue to fall short in the face of natural distribution shifts [31, 41]. Indeed, ERM has repeatedly shown to be a strong baseline – frequently outperforming dedicated methods that leverage domain information or additional unlabelled data – for DG [31, 63], despite the theoretical problems associated with using it when the training and test sets are misaligned. Until now, only pre-training on larger, more diverse datasets (with harder examples), has consistently proven to improve OOD generalisation, yet allowing pre-trained models to fit the ID data too closely can undo any such benefit conferred by the pre-training [3, 39, 69, 74]. Similar to Okapi, MatchDG [51] draws upon causal matching to tackle DG. Despite the surface-level similarity, there are a number of

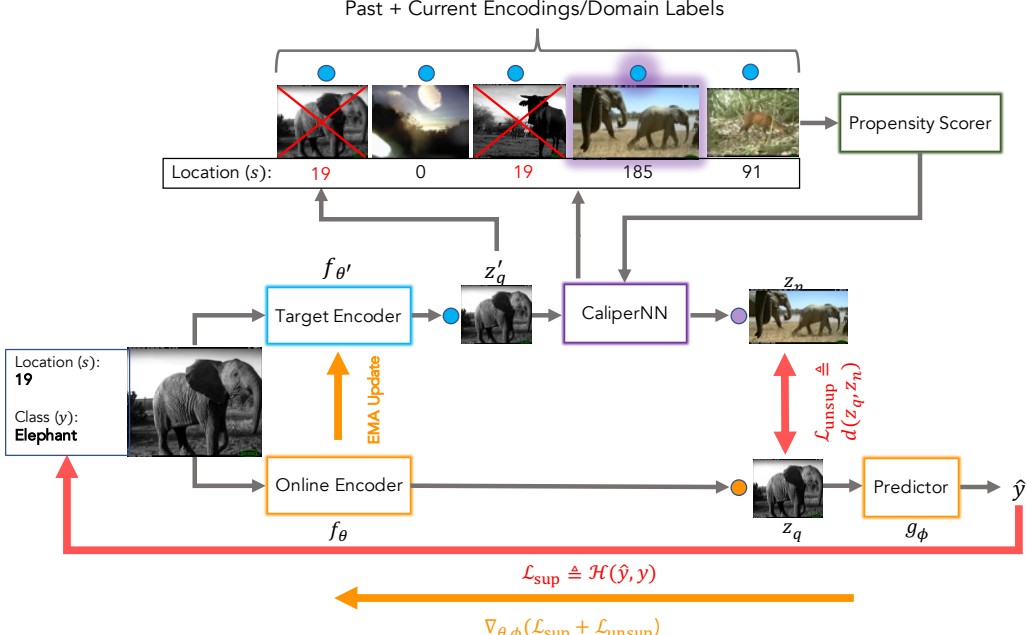

Figure 2: Overview of Okapi's online-learning pipeline based using the iWildCam dataset for the sake of illustration. For simplicity, we limit $k$ to 1 so that the output of matching is a single vector rather than a set of vectors; for the same reason we illustrate the process for only a single sample taken from the labelled data set $\mathcal{D}_l$, annotated with both domain ($s$; in this case, *camera location*) and class ($y$) information. Inspired by recent advances in self-supervised learning, we maintain a copy (the target encoder) of the online encoder, $f_\theta$, whose parameters, $\theta'$, are an exponential moving average (EMA) of $\theta$. This EMA update is performed at the beginning of each training set at a rate governed by the decay coefficient, $\zeta$. For a given sample, we first compute its embedding using the target encoder to produce the query vector, $z_q'$, and by the online encoder to produce $z_q$, which will serve as the 'anchor' in the consistency loss. This query vector is then used – alongside the output of the propensity scorer – by CaliperNN to compute its cross-domain nearest neighbour, $z_n$, where the keys are taken to be the current and past (stored in the Memory Bank) $N_\mathcal{M}$ encodings of the data. The cross-domain constraint, prohibiting matching of samples belonging to the same domain, is denoted through a red coloring of the location identifiers, the nearest sample obeying this constraint and the constraints of the calipers with purple highlighting. The consistency loss is the distance between $z_q$ and $z_n$, defined by function some distance function $d$. Finally, the supervised loss, $\mathcal{L}_{\text{sup}}$ (here instantiated as the standard cross-entropy loss, $\mathcal{H}$), is computed using the output of the predictor acting on $z_q$ and the ground-truth given by $y$.

significant differences, principally in the respects that we consider semi-supervised DG (whereas MatchDG requires full-labeling w.r.t. $y$) and employ an augmented form of k-NN for bias-reduction in the absence of $y$.

**Self-Supervised Learning**    In self-supervised learning (SelfSL), models are trained to solve pretext tasks constructed from the input data. This learning paradigm has led to significant breakthroughs in unsupervised learning in recent years, with performance now approaching (or even surpassing, along some axes such as adversarial robustness) that of supervised methods for many tasks while requiring significantly less labelled data. Due to its generality, SelfSL has seen use across the complete spectrum of applications and modalities and underlies many of the foundation models [11] that have emerged in NLP [13, 16, 21], Computer Vision [29], and at their intersection [2, 78]. Common pretext tasks include those based on the masked-language-modelling approach – originally popularised by BERT [21] and recently generalised to other modalities [6, 7] – [15, 33], contrastive captioning [56, 78], and instance discrimination and self-distillation [14, 30] which rely on transformations of the data to generate multi-view inputs. Approaches belonging to the latter two categories were originally limited by the fact that the transforms had to be tailored for a particular modality and for

Table 1: A comparison between Okapi and different baselines on two benchmark image datasets. We include both the results of our re-run of the baselines and those of [63]. Both ID and OOD performances are reported. For iWildCam we average over results from 3 different seeds, for PovertyMap we do so over the 5 pre-defined CV folds. Standard deviations are shown in parentheses.

| Method | iWildCam | | PovertyMap | | | |
| | macro F1 ↑ | | worst U/R corr. ↑ | | worst U/R MSE ↓ | |
| | ID | OOD | ID | OOD | ID | OOD |
|---|---|---|---|---|---|---|
| ERM [63] | 47.0 (1.4) | 32.2 (1.2) | 0.66 (0.04) | 0.49 (0.06) | - | - |
| FixMatch [63] | 46.3 (0.5) | 31.0 (1.3) | 0.54 (0.10) | 0.30 (0.11) | - | - |
| ERM | 48.6 (1.1) | 33.3 (0.3) | 0.72 (0.03) | 0.53 (0.09) | 0.23 (0.03) | 0.35 (0.12) |
| FixMatch | 51.1 (1.0) | 35.2 (0.7) | 0.50 (0.13) | 0.34 (0.12) | 0.59 (0.42) | 0.88 (0.61) |
| Okapi (ours) | 50.6 (0.7) | 36.1 (0.9) | 0.72 (0.02) | 0.55 (0.10) | 0.22 (0.02) | 0.33 (0.10) |
| Okapi (no calipers) | - | - | 0.72 (0.02) | 0.54 (0.12) | 0.22 (0.02) | 0.36 (0.14) |

some modalities, such as tabular data, there is no obvious way to define them. A number of recent works have sought to obviate this problem through the use of MixUp [72], masking [6, 34], and k-NN [24, 42, 71], the latter of which is directly relevant to our work. Okapi bears closest resemblance to [42] in combining momentum-encoding with nearest-neighbours lookup to generate the views for a BYOL-style [30] consistency loss. However, a key distinction lies in the use of an augmented form of nearest-neighbours, CaliperNN, which both constrains pairs of samples to being from *different* domains and filters out any queries or keys deemed outliers according to a learned *propensity score*.

**Semi-Supervised Learning** Semi-supervised learning (SemiSL) encompasses a broad class of algorithms that combine unsupervised learning with supervised learning in order to improve the performance of the latter, especially when labelled data is limited. Many SemiSL methods are based on the self-training paradigm which can trace its roots back decades to the early work in pattern recognition by [65] and continues to be relevant in the modern era due to its generality, both within SemiSL itself and in related fields such as domain adaptation [27], and fledgling field of SelfSL [14] discussed above. Self-training applies to any framework predicated on using a model's own predictions to produce pseudo-labels for the unlabelled data which can either be used as targets for self-distillation [75] or enforcing consistency between predictions that themselves have been perturbed [5, 75] or that have been generated from perturbed/multi-view inputs [67]. FixMatch [67] is one example of a consistency-based method which has proven effective for semi-supervised classification, despite its simplicity, and various works [28, 46] have since built on the its framework prescribing the use of weakly- and strongly-augmented inputs to generate the targets and predictions, respectively. Like these methods, Okapi also makes use of a cross-view consistency loss, however, the alternative views for a given sample are generated not through data-augmentation but through statistical matching [58], with the aim being to achieve invariance to the domain rather than a particular series of perturbations. Another example of particular relevance to our work is [70], which uses a copy of the model with exponentially-averaged weights to generate the targets for the unlabelled data. Okapi also uses such a model to produce the targets for its consistency loss, but is more akin to momentum-encoding [33] in the respect that the loss is imposed on the latent space.

## 5 Experiments

### 5.1 Datasets

We evaluate Okapi on three datasets taken from the WILDS 2.0 benchmark [63]. These span a variety of modalities and tasks, allowing us to showcase the generality of our proposed method (Okapi): **iWildCam** (images, multiclass classification), **PovertyMap** (multispectral images, regression), and **CivilComments** (text, binary classification). Details of each dataset can be found in Appendix A.

### 5.2 Image experiments

Results of our image-data experiments are summarised in Table 1. Due to spacial constraints, we defer the full set of results, including those for the 'offline' (w.r.t. the matching) version of Okapi to Appendix. C. For both datasets in question, we use the same metrics as [63]: macro-F1 for iWildCam

and worst-group (with the group defined as urban (U) vs. rural (R)) Pearson correlation for Poverty Map. For completeness, we include mean squared error (MSE) as a secondary metric for the latter dataset. Following [63], we compute the mean and standard deviation (shown in parentheses) over multiple runs for both ID and OOD test sets, with these runs conducted with 3 different random seeds and 5 pre-defined cross-validation folds for iWildCam and PovertyMap, respectively.

We compare Okapi against two baselines, ERM and FixMatch [67], both according to our re-implementation and according to the original implementation given in [63]. We note that since FixMatch, in its original form, is only applicable to classification problems due to its use of confidence-based thresholding, for the PovertyMap dataset, FixMatch represents a simplified variant (following [63]) without such thresholding, that is trained to simply minimise the MSE between *all* regressed values for the weakly- and strongly-augmented images. As described in Appendix D, the main difference between the baselines run included in [63] and our re-runs is in the backbone architecture, with us opting for a ConvNeXt [47] architecture over a ResNet one. For both datasets, and for both baselines we observe significant improvements stemming the change of backbone. Moreover, utilising ConvNeXt seems to be crucial in enabling FixMatch to surpass the ERM baseline in the classification task with 32.2 (ERM) vs 31.0 (FixMatch) and 33.3 (ERM) vs. 35.2 (FixMatch), with ResNet and ConvNeXt architecture respectively.

Okapi, convincingly outperforms the baselines, w.r.t the OOD metric of interest, on both datasets. We observe an improvement of +0.9 macro F1, i.e. 36.1 vs 35.2 of Okapi and FixMatch (the best baseline for iWildCam) respectively. For the regression task in PovertyMap, Okapi achieves 0.55 and 0.33 on the OOD test set in terms of Pearson correlation and MSE, respectively, in contrast to the 0.53 and 0.33 of ERM. At the same time, we note that FixMatch fails to generalise well to this task, yielding by far the worst results amongst the evaluated methods.

## 5.3 Text classification

| Method | Civil Comments worst-group acc ↑ |
|---|---|
| | OOD |
| ERM [63] | 66.6 (1.6) |
| ERM (fully-labelled) [63] | 69.4 (0.6) |
| ERM (reproduction) | 68.5 (2.2) |
| Okapi (ours) | 69.7 (2.0) |

Table 2: Comparison between Okapi and the baselines methods on the Civil Comments dataset. We include both the original results of [63] as well as those of our reproduction of their ERM baseline. Performance is measured in terms of worst-group accuracy and averaged over seeds; standard deviations are shown in parentheses.

In Table 2 we summarise the numerical results for the CivilComments dataset. Remaining consistent with [63], we evaluate models according to the worst-group accuracy – the minimum of the conditional accuracies obtained by conditioning on each of the 8 dimensions of $s$ – averaged over 5 replicates. Since there is no canonical ID test split available for this dataset, we report only the results only for the OOD split that is, rather than doing so for a custom split to avoid misrepresentation. We compare Okapi against both ERM variants featured in [63] – one trained on only the official labelled data and one trained with annotated unlabelled data (fully-labelled) – as well as our re-implementation of the ERM variant trained on only the labelled data with an identical hyperparameter configuration to the former. In contrast to the image datasets, we do not diverge in our choice of architecture, with all models trained with a pre-trained DistilBERT [64] backbone.

We observe marked improvement in the worst-group accuracy of this baseline compared with that reported therein. We attribute this partly to the high variance of the model-selection procedure (inherited from [63]) based on intermittently-computed validation performance (which does not consistently align with test performance) to determine the final model. This aside, we observe that Okapi outperforms the ERM baseline by a significant margin, to the point of parity with the fully-labelled baseline.

## 5.4 Ablations and qualitatitive analysis

In order to evaluate the importance of the caliper-based filtering to the performance of Okapi, we perform an ablation experiment on PovertyMap dataset (Okapi (no calipers)) with said filtering disabled (and all else constant), such that instead of CaliperNN we have standard $k$-NN, albeit with

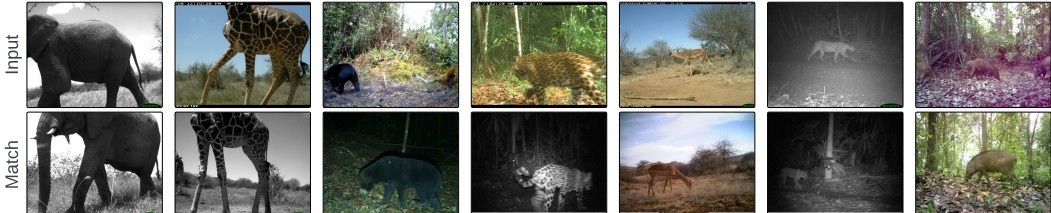

Figure 3: Examples of input (labelled) images and their 1-NN matched (unlabelled) images retrieved using CaliperNN on iWildCam dataset. Here, we match images from the labelled-train set to images from the unlabelled-extra set, taking advantage the fact that their domains are disjoint.

the cross-group constraint still in place (per Eq. 2). We see that performance degrades according to both metrics of interest, and, crucially, that the standard deviation of the runs is significantly higher, in line with our expectation that filtering out poor matches should stabilise optimisation. We provide additional ablation experiments in Appendix F, exploring the relative importance of the two (fixed and std-) calipers, the optimal number of neighbours to use for computing $\mathcal{L}_{\text{unsup}}$, and the feasibility of using the online encoder to generate the queries for CaliperNN.

Finally, in Fig. 3 we show samples of matched pairs retrieved by CaliperNN from the encodings of the learned encoder for the iWildCam dataset. Here, we see that semantic information (encoding the species of animal) is preserved across pairs, while nuisance factors such as illumination, background and contrast vary. Further examples from PovertyMap are shown in Appendix E. In Appendix H, we include matching results for the PACS (photo (P), art painting (A), cartoon (C), and sketch (S)) dataset [45] demonstrating how temperature scaling, in conjunction with the fixed caliper, can be used to control the filtering rate.

## 6  Conclusion

In this work, we introduced, Okapi, a semi-supervised method for training distributionally-robust models that is intuitive, effective, and is applicable to any modality or task. Okapi is based on the simple idea of supplementing the supervised loss with a cross-domain consistency loss that encourages the outputs of an encoder network to be similar for neighbouring (within the latent space of the encoder itself) samples belonging to different domains, which is made efficient using an online-learning framework. Rather than simply using $k$-NN with a cross-domain constraint, however, we propose an augmented form based on statistical matching (CaliperNN) that combines propensity scores with calipers to winnow out low-quality matches; we find this to be important for both the end-performance and consistency of Okapi. Our work serves as a response to [63], in that we find that it is in fact possible to effectively incorporate unlabelled data and domain information into a training algorithm in order to improve upon ERM with respect to an OOD test set, assuming an appropriate choice of architecture. Namely, on three datasets from the WILDS 2.0 benchmark, representing two different tasks (classification and regression) and modalities (image and text), we show that Okapi outperforms both the ERM and FixMatch baselines according to the relevant OOD metrics.

Buoyed by these promising results, we intend to apply Okapi to other tasks (e.g. object detection and image segmentation) and other modalities (e.g. audio) to further establish its generality. Furthermore, one limitation of the current incarnation of the method is that the thresholds for the calipers are fixed over the course of training whereas it may be beneficial to set these adaptively with the view to optimise such measures of inter-domain balance as *Variance Ratio* and *Standard Mean Differences* that are commonly used to evaluate the the goodness of statistical matching procedures.

## Acknowledgments and Disclosure of Funding

This research was supported by a European Research Council (ERC) Starting Grant for the project "Bayesian Models and Algorithms for Fairness and Transparency", funded under the European Union's Horizon 2020 Framework Programme (grant agreement no. 851538). NQ is also supported by the Basque Government through the BERC 2018-2021 program and by Spanish Ministry of Sciences, Innovation and Universities: BCAM Severo Ochoa accreditation SEV-2017-0718.

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
