# A  Datasets

We evaluate Okapi using three datasets – iWildCam, PovertyMap, and CivilComments – taken from the WILDS 2.0 benchmark [63]. These datasets were chosen specifically due to the poor performance reported by [63] for semi-supervised and domain adaptation methods across the board, in relation to the ERM baselines. For PovertyMap in particular, ERM was found to vastly outperform any competing methods utilising the unlabelled data and/or domain labels.

**iWildCam-WILDS** is an extension of the iWildCam 2020 Competition Dataset [8]. The task is multi-class species classification of animals in camera trap images. The dataset contains 1022K images of animals annotated with the domain, $s$, that identifies the camera trap that captured it. The target label, $y$, is one of 182 different animal species and it is provided solely for the 203K labelled data. The labelled training set contains 130K images taken by 243 camera traps. The out-of-distribution (OOD) validation and target sets include images from 32 and 48 different camera traps which are disjoint from the 243 training domains. Additionally, 819K unlabelled images from 3215 new domains are available. Different cameras trap differ in characteristics such as illumination, background and relative animal frequency, models trained on the source domains might fail to generalise to images taken from new locations.

**PovertyMap-WILDS** is a variation of the dataset introduced in [76]. The task is to predict the wealth index, $y$, from multispectral satellite images of 23 African countries. The country the image was taken in as well as whether it was taken in a rural or urban area represent the domain $s$. The dataset contains 5 cross-validation (CV) folds of roughly equal size, each one dividing the 23 countries differently across the source, OOD validation and OOD target splits. In each fold, the labelled training set contains 11K images from 14 different countries. The OOD validation and target sets include images from 5 different countries not represented in the source data. The dataset also includes 261K unlabelled images from the same 23 countries.

**CivilComments-WILDS** is an online-comment dataset adapted from [12], comprising 448K online comments annotated with both a binary indicator of toxicity ({$toxic, not toxic$}) – serving as the target label, $y$ – and the demographic identities mentioned within them – serving as the domain $s$. Here, $s \in \{0, 1\}^8$ is a binary vector rather than a scalar, with dimensions indicating membership (non-exclusively) to 8 demographic groups, spanning different genders, religions and ethnicities. For the WILDS 2.0 variant of the dataset, [63] introduce an additional corpus of 1551K comments acting as the unlabelled training data belonging to extra domains. While the comments are completely unlabelled, w.r.t. both $y$ and $s$ and thus are not domain-separable at the sample level, the majority (92%) of the comments are known to be sourced from the same documents as those comments comprising the (OOD) labelled test data. As noted in [63], CivilComments-WILDS exhibits label imbalance w.r.t. $y$; this is amended both therein and herein (as appertains all methods) through the use of class-balanced sampling, though with the minor distinction that for our experiments we ensure each batch is exactly balanced rather by sampling equally from each class, in contrast to [63] who sample hierarchically – sampling $y_i$ uniformly from $\{0, \ldots, |\mathcal{Y}_l|\}$ and then uniformly from $\mathcal{D}_l$, conditioned on $y_i$ – such that balance is achieved only in expectation.

# B  Relation to Algorithmic Fairness

DG and Algorithmic Fairness overlap in their objective to train a model that yields predictions that are statistically independent of (and thus robust to variations in) domain, when for the latter the domain is taken to be some protected characteristic, such as age or gender, and fairness is measured according to invariance-driven notions of group fairness such as Demographic Parity [26] and Equal Opportunity [32]. Indeed, methods that focus on equalising the empirical risk across subgroups – such as by importance weighting [36, 66] – have featured extensively in both DG [4, 19, 43, 62] and fairness [1, 22, 37] and many approaches to fair representation learning [18, 38, 50, 53, 55] have roots in the former [52] and in the closely-related field of domain adaptation [27]. Beyond this more general equivalence, our work also has ties to notions of individual fairness pioneered by [25] – broadly prescribing that similar individuals be treated similarly – in that our unsupervised loss involves maximising the similarity between inter-domain samples within representation space. This is reminiscent of the operationalisation of individual fairness proposed by [44] that enforces similarity between a given representation and the representations of its neighbouring – in both the input space and according to a between-group (cross-domain) quantile graph – samples.

Table 3: An extended comparison between Okapi and different baselines on two benchmark image datasets. We include both the results of our re-run of the baselines and those of [63]. Both ID and OOD performances are reported. For iWildCam we average over results from 3 different seeds, for PovertyMap we do so over the 5 pre-defined CV folds. Standard deviations are shown in parentheses. The additions relative to Table 1 include results with an offline variant of Okapi – where the matches are generated prior to training from the features of the trained ERM model and then fixed for the course of training – and with the ResNet backbones employed by [63].

| Method | iWildCam | | PovertyMap | | | |
| | macro F1 ↑ | | worst U/R corr. ↑ | | worst U/R MSE ↓ | |
| | ID | OOD | ID | OOD | ID | OOD |
|---|---|---|---|---|---|---|
| ERM [63] | 47.0 (1.4) | 32.2 (1.2) | 0.66 (0.04) | 0.49 (0.06) | - | - |
| FixMatch [63] | 46.3 (0.50) | 31.0 (1.3) | 0.54 (0.10) | 0.30 (0.11) | - | - |
| ERM (ConvNeXt) | 48.6 (1.1) | 33.3 (0.3) | 0.72 (0.03) | 0.53 (0.09) | 0.23 (0.03) | 0.35 (0.12) |
| FixMatch (ConvNeXt) | 51.1 (1.0) | 35.2 (0.7) | 0.50 (0.13) | 0.34 (0.12) | 0.59 (0.42) | 0.88 (0.61) |
| Okapi (ours; ConvNeXt) | 50.6 (0.7) | 36.1 (0.9) | 0.72 (0.02) | 0.55 (0.10) | 0.22 (0.02) | 0.33 (0.10) |
| Okapi (offline; ConvNeXt) | 48.8 (0.8) | 31.7 (0.2) | 0.68 (0.02) | 0.53 (0.07) | 0.26 (0.02) | 0.37 (0.13) |
| Okapi (no calipers; ConvNeXt) | - | - | 0.72 (0.02) | 0.54 (0.12) | 0.22 (0.02) | 0.36 (0.14) |
| ERM (ResNet) | 46.5 (0.8) | 29.7 (1.0) | 0.69 (0.03) | 0.53 (0.08) | 0.24 (0.04) | 0.34 (0.11) |
| FixMatch (ResNet) | 43.0 (2.5) | 25.5 (1.4) | 0.70 (0.02) | 0.53 (0.08) | 0.24 (0.02) | 0.35 (0.10) |
| Okapi (ResNet) | 46.1 (0.7) | 27.8 (0.3) | 0.70 (0.04) | 0.52 (0.07) | 0.23 (0.02) | 0.33 (0.10) |

## C  Extended Results

We tabulate in Table 3 an extended version of the results presented in the main text. This includes additional results with the ResNet backbones per [63] (justifying our decision to adopt a ConvNext backbone for our main set of image-dataset results) as well as those for an 'offline' version of Okapi (Okapi (offline)) where the matches are generated prior to training using features of the respective ERM baseline for each dataset. Since the target encoder is necessitated by the need for online match-retrieval, only a single encoder is involved in Okapi (offline); in binary cases, the algorithm is then identical to the one proposed by [57] with the exception that consistency is still enforced via distance in encoding space rather than with a JSD loss on the predictive distributions which fails to generalise to regression tasks such as PovertyMap.

## D  Implementation details

**Data Augmentation**  We follow [63] when defining the augmentations for the the WILDS datasets. In the case of PovertyMap-WILDS we corroborate the original finding that data-augmentation adversely affects performance, and, in light of this, elect only to use data-augmentation for FixMatch where it is needed to generate the weak and strong views used in computing the consistency loss. Since Okapi uses an NN-based approach for generating these views, it is decoupled from the augmentation strategy and problems that can arise from its misspecification.

**Architecture**  For our image experiments, contrary to [63], we opt to use the recently proposed ConvNeXt architecture [47], finding this change to provide large performance gains and to be crucial in enabling semi-supervised methods to surpass the ERM baseline. This is in line with [39] who similarly found that a change of architecture (combined with large-scale pre-training) could greatly bolster performance on the iWildCam dataset. More precisely, we use the *tiny* variant of ConvNeXt, pre-trained on ImageNet 1k, as the initial backbone for our models. We compose this with a single fully-connected layer to construct the complete predictor both for the target and the propensity score. For our CivilComments experiments, in contrast, we do not diverge from [63] in our choice of architecture, with all models trained with a pre-trained DistilBERT [64].

**Optimisation**  For optimising all models, we use the AdamW optimiser [49] coupled with a cosine annealing schedule without warm restarts [48]. We set the initial learning to be $1 \times 10^{-4}$ across the board, and forgo the use of weight decay. Models are trained for 120K, 30K, and 20K iterations for iWildCam, PovertyMap, and CivilComments, respectively. The decay coefficient, $\zeta$, for the target encoder's exponential moving average is initialised to $\zeta_{start}$ and is linearly increased to $\zeta_{end}$ over the course of training. For PovertyMap we set $\zeta_{start}$ and $\zeta_{end}$ to be 0.996 and 1.0, respectively; for iWildCam, we set $\zeta_{start}$ and $\zeta_{end}$ to be 0.999 and 0.999, respectively, resulting in a fixed value of

$\zeta$; 1.0, respectively; for CivilComments, we set $\zeta_{\text{start}}$ and $\zeta_{\text{end}}$ to be 0.996 and 0.996, respectively, again resulting in a fixed value of $\zeta$. We similarly warm up the pre-factor for the consistency loss, $\lambda$, according to a linear schedule during the first 10% of training to allow a period for the encoder to learn meaningful relations between samples through the supervised loss before bootstrapping with the consistency loss, with a final value of 1.

**Matching**    In order to determine suitable hyperparameters, $\xi$, for CaliperNN, we perform a grid-search in the static setting, using a fixed model. Specifically, we use the backbone of an ERM-trained model as the encoder with which to generate the queries and keys for matching. The quality of matching with a given instantiation of $\xi$ is measured using two metrics commonly used in the statistical matching literature: *Variance Ratio* (VR) and *Standard Mean Differences* (SMD) [61]. Both metrics operate on pairs of domains, but can be generalised to work when $s$ is non-binary by simply aggregating over all pairwise results. For a given pair of domains, VR is defined as the ratio of the variances of the covariates between the two domains, with an ideal value of 1, while SMD is defined as the difference in their covariate means, normalised by the standard deviation for each covariate, and is to be minimised. While our proposed method is applicable whether $S$ is binary or categorical, for the experiments in this paper we take advantage of the fact that the WILDS datasets specify splits with non-overlapping domains and match from $\mathcal{D}_l \to \mathcal{D}_u$ and in the reverse direction (from $\mathcal{D}_u \to \mathcal{D}_l$). This decision was based on preliminary experiments which found the binary variant generally enjoyed more stable optimisation, something which future work should seek to rectify. In the case of PovertyMap, however, the training splits themselves do not satisfy the aforementioned requirement of being sourced from mutually exclusive sets of domains and we instead treat the OOD validation set as $\mathcal{D}_u$ (and treat it as being unlabelled w.r.t. $y$, in that it is only used for $\mathcal{L}_{\text{unsup}}$).

# E    Additional Matching Examples

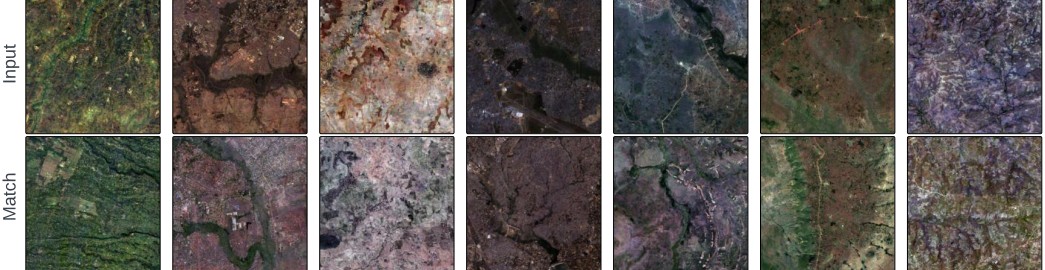

Figure 4: Examples of input (labelled) images and their 1-NN matched (unlabelled) images retrieved using CaliperNN from the PovertyMap-WILDS dataset. Here, we match images from the labelled-train set to images from the OOD-validation set, taking advantage the fact that their domains are disjoint.

# F    Ablations

We supplement the ablation experiment on the use of calipers featured in the main text with additional experiments concerning effect of the number of nearest neighbours ($k$), the relative importance of the two (fixed and std) calipers, and the feasibility of using the online encoder as the query-generator instead of the target encoder. The results of these experiments are tabulated in 4 with the key takeaways being:

1. The number of neighbours used for computing the consistency loss has little impact – according to the given level of precision – on the performance of Okapi along all axes.

2. While disabling the calipers altogether considerably harmed performance, using only the std. caliper allows us to recover the performance of the complete algorithm, (`Okapi (k=5)`), whereas the same is not true for the fixed caliper which, while aiding performance compared to the no-caliper baseline, falls short of that benchmark. A caveat attached to these conclusions, however, is that the selected values for $\xi$ are likely suboptimal in the online

Table 4: Ablation experiments for Okapi conducted using the PovertyMap-WILDS dataset. Specifically, we assess the importance of four elements of our proposed method: the number of nearest neighbours used in computing $\mathcal{L}_{\text{unsup}}$ ($k$), the use (enabled/disabled) of the fixed- and std-calipers in CaliperNN (considering these to be separate components), and which encoder (online or target) is used to generate the queries for statistical matching (with use of the target encoder 'TE queries' being the default and 'OE queries' denoting the alternative). Both ID and OOD performances are reported. The results are computed by aggregating over the results for each of the 5 pre-defined cross-validation folds. We report the average and standard deviation value across replicates of the metric of interest.

(a) CaliperNN ablations.

| Method | PovertyMap | | | |
| --- | --- | --- | --- | --- |
| | worst U/R corr. ↑ | | worst U/R MSE ↓ | |
| | ID | OOD | ID | OOD |
| Okapi (k=1) | 0.72 (0.02) | 0.55 (0.10) | 0.22 (0.02) | 0.33 (0.10) |
| Okapi (k=5) | 0.72 (0.02) | 0.55 (0.10) | 0.22 (0.02) | 0.33 (0.10) |
| Okapi (k=10) | 0.72 (0.02) | 0.55 (0.09) | 0.22 (0.02) | 0.33 (0.10) |
| Okapi (k=5, no calipers) | 0.72 (0.02) | 0.54 (0.12) | 0.22 (0.02) | 0.36 (0.14) |
| Okapi (k=5, no std caliper) | 0.72 (0.02) | 0.54 (0.12) | 0.22 (0.02) | 0.35 (0.14) |
| Okapi (k=5, no fixed caliper) | 0.72 (0.02) | 0.55 (0.10) | 0.22 (0.02) | 0.33 (0.10) |

(b) Query-generator ablation (target encoder (TE) vs. online encoder (OE)).

| Method | PovertyMap | | | |
| --- | --- | --- | --- | --- |
| | worst U/R corr. ↑ | | worst U/R MSE ↓ | |
| | ID | OOD | ID | OOD |
| Okapi (TE queries) | 0.72 (0.02) | 0.55 (0.10) | 0.22 (0.02) | 0.33 (0.10) |
| Okapi (OE queries) | 0.72 (0.02) | 0.55 (0.10) | 0.23 (0.02) | 0.34 (0.10) |

setting, given that they were optimised for the static setting: with improved selection of $\xi$, either by learning it jointly with the model's parameters (using, for instance, the perturbed maximum method [9] to overcome the non-differentiability of the $k$-NN and thresholding operations), in an amortised fashion, or optimising it on a per-iteration basis.

3. While less appealing from a conceptual standpoint, due to the mismatch between the networks used to generate the queries and keys, from an empirical standpoint it is perfectly feasible to use the online encoder to generate the queries for statistical matching instead the target encoder while experiencing minimal degradation in performance. This is particularly relevant when one wishes to perform the matching in only one direction (e.g. $\mathcal{D}_l \to \mathcal{D}_l$) due to the reduction in redundant encoding, with each encoder only encoding its respective subset of the data (e.g. $f_\theta$ only encodes samples from $\mathcal{D}_l$ and $f'_\theta$ only encodes samples from $\mathcal{D}_u$)

# G  Pseudocode

We provide Pytorch-style [54] pseudocode for the CaliperNN (described in 3.2) and online-learning (described in 3.3) algorithms in Algorithm 1 and Algorithm 2, respectively. In both cases, we restrict the pseudocode to the special case of binary domains – practically achieved by using the labelled/unlabelled as a proxy for domain – for ease of illustration. The CaliperNN algorithm can be generalised freely to multiclass cases by considering pairwise interactions between the propensity scores for each domain for applying the calipers and by computing the pairwise inequalities between $\mathbf{s}_q$ and $\mathbf{s}_k$ (giving the connectivity matrix $(\mathbf{s}_q \cdot \mathbf{1}^T) \neq (\mathbf{1} \cdot \mathbf{s}_k^T)$, where $\mathbf{1}$ denotes the ones vector of the same shape as its multiplicand and mediates broadcasting) for enforcing the cross-domain constraint.

**Algorithm 1:** Pytorch-style pseudocode for the CaliperNN matching algorithm for the special case where the domain is binary. The algorithm generalises freely to arbitrary numbers of domains however we restrict ourselves to the binary version here for illustrative purposes.

```python
def binary_caliper_nn(
    x_query, # samples to be used as the queries for matching
    s_query, # binary labels indicating the domain of x_query
    x_key, # samples to which the query samples may be matched to.
    s_key, # binary labels indicating the domain of x_key
    ps_query, # propensity scores associated with x_query
    ps_key, # propensity scores associated with x_key
    t_f, # threshold for the fixed caliper
    t_sigma, # number of standard deviations at which to threshold
    k # number of neighbours to attempt to retrieve per query
):
    anchor_inds, positive_inds = [], []
    for direction in (0, 1): # which domain (0 or 1) to treat as the 'anchor'
        key_mask = s_key != direction
        # exclude samples with propensity scores outside the valid range
        # determined by t_f: (1 - t_f, t_f)
        fc_mask = (ps_query > (1 - t_f)) &  (ps_query < t_f)
        anchor_mask = fc_mask & (s_query == direction)
        queries_x_filtered = queries.x[anchor_mask]
        ps_query_filtered = ps_query[anchor_mask]
        fc_mask = (ps_key > (1 - t_f)) & (ps_key < t_f)
        key_mask &= fc_mask
        ps_key_filtered = ps_key[key_mask]
        # 2-norm distance between unfiltered propensity scores
        dists_ps = cdist(ps_query_filtered, ps_key_filtered, p=2)
        # 2-norm distance between the filtered anchors and keys
        dists_x = cdist(queries_x_filtered, x_key[key_mask], p=2)
        # compute sigma as the mean of the per-domain standard deviations
        std_ps = (0.5 * (ps_query_filtered.var() + ps_key_filtered.var())).sqrt()
        std_threshold = t_sigma * std_ps
        # filter out any samples that violate the std-caliper
        dists_x[dists_ps > std_threshold] = float("inf")
        nbr_dists, nbr_inds = dists_x.topk(dim=1, largest=False, k=k)
        # filter out queries not yielding the requisite number of matches (k)
        is_matched = ~nbr_dists.isinf().any(dim=1)

        anchor_inds.append(anchor_mask.nonzero()[is_matched])
        positive_inds.append(key_mask.nonzero()[nbr_inds[is_matched]])

    return cat(anchor_inds, dim=0), cat(positive_inds, dim=0)
```

# H  Matching for PACS dataset

In this section we perform some initial experiments on PACS dataset [45] (using features extracted for a pre-trained CLIP [56] model) to show how the temperature scaling can be used to smooth the propensity score distribution to better control how many sample are discarded during matching. There are 1,670 *photo*, 2,048 *art painting*, 2,344 *cartoon*, and 3,929 *sketch* in the dataset. Here will evaluate the results of matching across the two domains *photo* and *art painting* as well as across *photo* and *sketch*. In Fig. 5 and Fig. 6 we compare the shape of the estimated propensity score with its scaled version using a temperature value of 10. As we can see, in the case of a distribution with extremely heavy tails (photo, sketch), the effect of smoothing the distribution is that when a fixed caliper is applied most of the samples are retained. On the other hand, when the initial distribution is

smoother, a temperature of 10 is extreme, having the effect of transforming the bimodal distribution to a unimodal one. Additionally, we tabulate in Table 5 the number of matched pairs retrieved when matching across the two domains photo and sketch; here we can see that by increasing the temperature we smooth the estimated propensity score distribution and thereby retain more samples. Similarly, we can retrieve more pairs by reducing the fixed caliper threshold. We also analyse the case of matching across the two domains photo and art painting. Using a fixed caliper defined defined by a threshold $t_f = 0.1$ and no temperature scaling (i.e. $\tau = 1$) the algorithm retrieves 1,142 pairs matching in the direction photo $\rightarrow$ art and 1,501 in the direction art $\rightarrow$ photo.

In Fig. 7 and Fig. 8 we show examples of matching pairs found using our CaliperNN algorithm. Although the features were not fine-tuned on PACS, we can see a few examples of intraclass matching. For the photo-art painting application we can see preservation in colour and background; while in the photo-sketch case shape and pose.

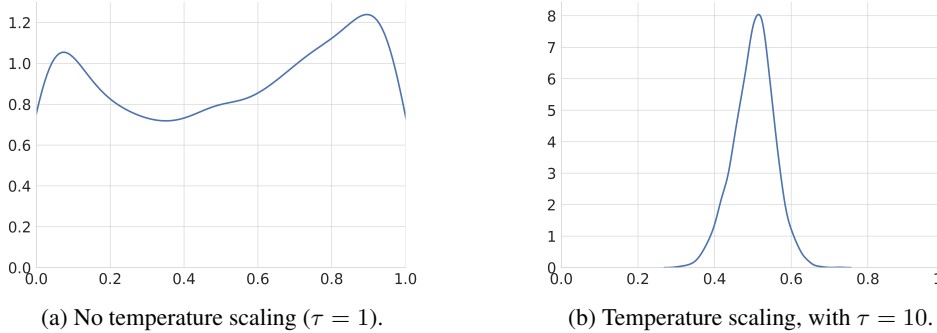

(a) No temperature scaling ($\tau = 1$).    (b) Temperature scaling, with $\tau = 10$.

Figure 5: Estimated propensity score distribution of *photo* and *art painting* on the PACS dataset. We compare (a) the original distribution ($\tau = 1$) and (b) the temperature-scaled distribution ($\tau = 10$). Here, the large temperature has the effect of transforming a bimodal distribution into a unimodal one.

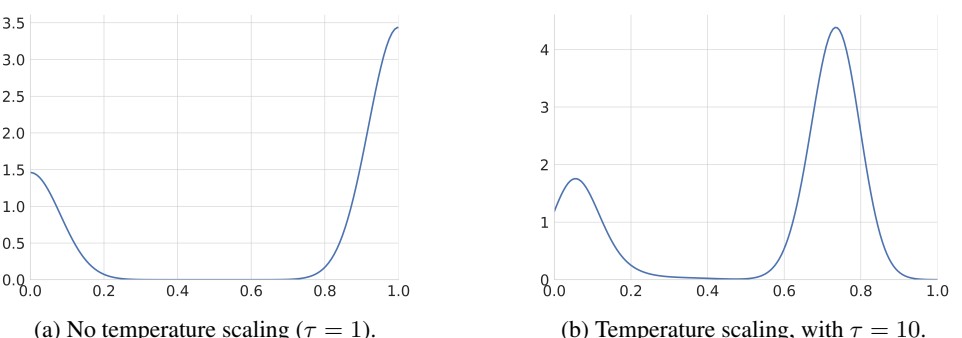

(a) No temperature scaling ($\tau = 1$).    (b) Temperature scaling, with $\tau = 10$.

Figure 6: Estimated propensity score distribution of *photo* and *sketch* on the PACS dataset. We compare (a) the original distribution ($\tau = 1$) and (b) the temperature-scaled distribution ($\tau = 10$). Here, the large temperature has the effect of smoothing the distribution.

# I  Energy and Carbon Footprint Estimates

To highlight the efficiency of Okapi, we provide estimates in 6 of the carbon footprint associated with the running of it and of the ERM and FixMatch baselines on the iWildCam dataset, using the same hyperparameter configuration used to generate the results in the main text. The runs were conducted in a controlled fashion, using the computing infrastructure and device count in all cases.

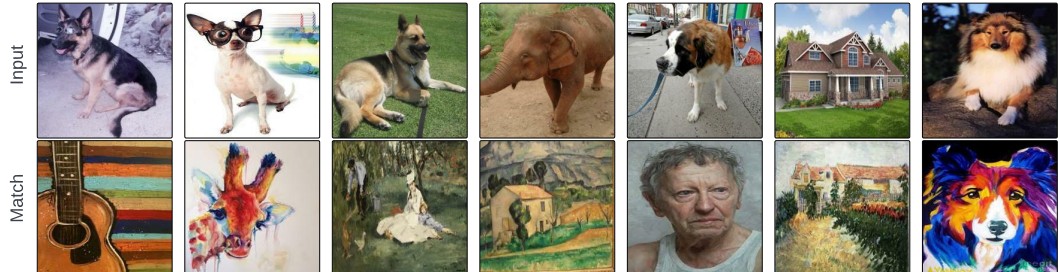

Figure 7: Examples of input (photo) images and their 1-NN matched (art paint) images retrieved using CaliperNN from the PACS dataset.

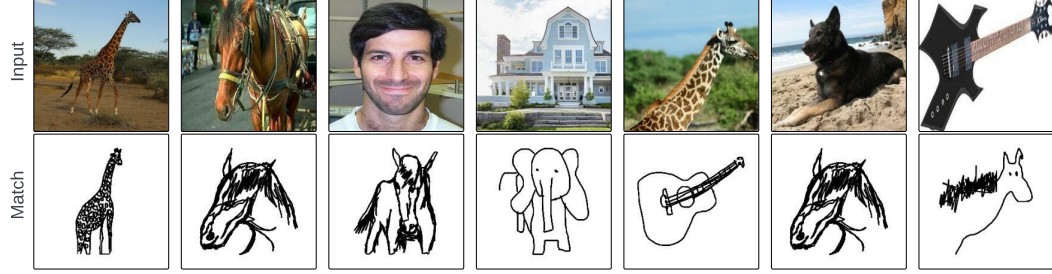

Figure 8: Examples of input (photo) images and their 1-NN matched (sketch) images retrieved using CaliperNN from the PACS dataset.

**Algorithm 2:** Pytorch-style pseudocode for the online learning algorithm for the special case where the labelled and unlabelled datasets are treated as the domains. The algorithm generalises freely to arbitrary numbers of domains however we restrict ourselves to the binary version here for illustrative purposes.

```python
# online_encoder: online encoder
# predictor_head: online predictor head
# propensity_scorer: online propensity scorer
# target_encoder momentum encoder (frozen)
# n_m: memory-bank capacity
# zeta: decay rate of the EMA updates
# tau: temperature-scaling parameter for the propensity scores.
# t_f: fixed caliper threshold for CaliperNN
# t_sigma: number of standard deviations at which to threshold in CaliperNN
# l_sup: supervised loss function
# k: number of matches to retrieve per query
# lambda_: loss pre-factor for the unsupervised loss
# D: Dimensionality of the encodings.

feature_mb = empty(n_m, D) # memory bank storing momentum-encoded features
label_mb = empty(n_m) # memory bank storing domain labels associated with feature_mb
 # load minibatches with B_l labelled samples and B_u unlabelled samples
for x_l, y, x_u in train_loader:
    # EMA update: \theta^\prime_t = \zeta \theta^\prime_{t - 1} + (1 - \zeta) \theta_t
    ema_update(target_encoder, online_encoder, zeta)
    features_o_l = online_encoder(x_l) # f_\theta(x_l) -> z_l
    features_t = target_encoder(cat([x_l, x_u])) # f_\theta(x_l \cup x_u) -> z_q^\prime
    y_hat = predictor_head(features_o_l) # g_\phi(z_l) -> \hat{y}
    features_o_u = online_encoder(x_u) # f(x_u) -> z_u
    features_o = cat([features_o_l, features_o_u]) # z_q := z_l \cup z_u
    # normalize the encodings to unit vectors.
    features_o_n = normalize(features_o, p=2, dim=1)
    queries = normalize(features_t, p=2, dim=1)
    # we treat x_l and x_u as coming from domains indexed by 0 and 1, respectively
    labels_l_q = ones(len(x_l)) # ones-vector of size B_l
    labels_u_q = zeros(len(x_u)) # zeros-vector of size B_u
    labels_q = cat([labels_l_q, labels_u_q])
    mb_mask = is_empty(label_mb) # mask indicating which elements of the MB are filled
    labels_k = cat([labels_q, label_mb[mb_mask].clone()])
    # keys are the union of the queries and the memory-bank-stored features
    keys = cat((queries, feature_mb[mb_mask].clone()), dim=0)
    feature_mb.push(queries) # update the feature memory bank
    label_mb.push(labels_q) # update the label memory bank
    logits_ps_k = propensity_scorer(keys) # h_\psi(z_k) -> e_k
    loss_ps = xent(logits_ps_k, labels_k) # (binary) cross-entropy loss
    # tempered logistic function: 1 / (1 + exp(-logits_ps_k / tau))
    logits_ps_k = sigmoid(logits_ps_k / tau)
    logits_ps_q = logits_ps_k[:len(queries)]
    # filter and match queries with (binary) CaliperNN
    inds_a, inds_p = binary_caliper_nn(
        features_t_n, labels_q, keys, labels_k,
        logits_ps_q, logits_ps_k, t_f, t_sigma, k
    )
    # compute the unsupervised loss (d(z_q, z_n)) for all matched queries
    z_q, v_k = features_o[inds_a], keys[inds_p]
    match_rate = len(z_q) / len(features_o)
    loss_u = match_rate * (z_q.unsqueeze(1) - v_q).pow(2).sum(-1).mean()
    loss = l_sup(y_hat, y) + lambda_ * loss_u + loss_ps # aggregate loss
    loss.backward() # compute gradients
    update(online_encoder, predictor_head, propensity_scorer) # optimizer updates
```

Table 5: Analysis of the number of the retrieved matched pairs when matching across the two domain *photo* and *sketch* on the PACS dataset. The fixed caliper threshold and temperature scaling can be used to smooth the propensity score distribution and effect the number of pairs.

| Fixed Caliper ($t_f$) | Temperature ($\tau$) | photo $\rightarrow$ sketch | sketch $\rightarrow$ photo |
|---|---|---|---|
| 0.1 | 1 | 0 | 0 |
| 0 | 1 | 1540 | 3929 |
| 0.01 | 1 | 6 | 9 |
| 0.01 | 1.3 | 14 | 56 |
| 0.01 | 1.8 | 25 | 574 |
| 0.01 | 2.5 | 41 | 3082 |
| 0.01 | 10 | 1540 | 3929 |
| 0.1 | 10 | 298 | 3929 |

Table 6: Comparison of the estimated carbon footprint (kgCoeq) of Okapi with the ERM and FixMatch baselines per replicate of the iWildCam dataset. For the controlled training conducted to enable fair computation of these estimates, we used a private infrastructure with an estimated carbon efficiency of 0.432 kgCOeq/kWh and RTX 3090 GPUs, each job being run on a single GPU, coupled with four data-loading workers.

| Method | kgCOeq $\downarrow$ |
|---|---|
| ERM | 1.36 |
| FixMatch | 2.12 |
| Okapi (ours) | 1.97 |