# OpenReview forum: "Okapi: Generalising Better by Making Statistical Matches Match"
_NeurIPS.cc/2022/Conference — NeurIPS 2022 Accept_

### Official Review · Reviewer_SyKB · 2022-07-04

**Rating:** 3
**Confidence:** 5
**Soundness:** 2 fair
**Presentation:** 1 poor
**Contribution:** 2 fair

**Summary:**

The authors studied semi-supervised learning across multiple domains. For each input example, they search for its k-nearest neighbors from other domains and enforce their feature representations to be similar. To filter outliers, the authors develop cross-domain matching based on the propensity score.

The authors applied the proposed algorithm to two datasets: iWildCam and PovertyMap. Empirical results show that their proposed method outperform previous baselines.

**Questions:**

In the algorithm, you tried to push the representations of the query example to be close to its unalbeled neighbors. Why would it help generalization? Aren't their representations already very close (since they are selected by k-NN)?

**Ethics Review Area:**

["I don’t know"]

**Limitations:**

No. The authors claimed that there is no negative societal impacts.

All papers have limitations. I think it is important for researchers to discuss the weakness of their proposed method so that people can better position the paper.

**Strengths And Weaknesses:**

Strengths:
1. The idea of enforcing feature consistency on the unlabeled data is interesting.


Weaknesses:
1. The empirical results (OOD 36.1 vs. 35.2) are far from satisfying. Moreover, the in-domain performance of the proposed method drops from 51.1 to 50.6. With 819K unlabeled examples (almost 8 times the size of the labeled data), I don't feel that this matching algorithm will be practically useful.
2. The presentation quality needs to be improved.
    + The introduction focused too much on those common sense knowledge and only talked about their proposed method in the last paragraph.
    + Confusing notation. For example in Eq (2), what are the arguments of $NN$? In Eq (4), what is $\text{CaliperNN}_\xi(z, \mathbf{z}, \mathbf{e}, s, k))$?
    + Given the complexity of the algorithm, I think the authors should include a pseudo-code in the main text.

---

> ### Author Response · Authors · 2022-08-02
> **Response to review**
>
> **The empirical results (OOD 36.1 vs. 35.2) are far from satisfying. Moreover, the in-domain performance of the proposed method drops from 51.1 to 50.6. With 819K unlabeled examples (almost 8 times the size of the labeled data), I don't feel that this matching algorithm will be practically useful.**
>
> It is true that for both datasets our method affords only a slight improvement with respect to the 'best baseline', however which baseline qualifies as such varies between iWildCam and PovertyMap: while FixMatch outperforms ERM on iWildCam, this relationship is reversed for PovertyMap. With Okapi we can improve the OOD F1 by +0.9 with respect to FixMatch (the best baseline) but +2.8 compared to ERM for iWildCam. FixMatch performs poorly when applied to PovertyMap while the improvement of our Okapi is marginal compared to the best baseline ERM (+0.05 OOD correlation and same OOD MSE) but significant compared to FixMatch (+0.21 OOD correlation and -0.55 OOD MSE).
>
> **The presentation quality needs to be improved. The introduction focused too much on those common sense knowledge and only talked about their proposed method in the last paragraph.**
>
> We agree that the first half of the introduction is overly general should be tailored to be more specific to the particular problem we tackle; we can assure the reviewer that this will be amended in the camera-ready version of the paper.
>
> **Confusing notation. For example in Eq (2), what are the arguments of $NN$ In Eq (4), what is $\mathrm{CaliperNN}(z, \mathbf{z}, \mathbf{e}, s, k)$? Given the complexity of the algorithm, I think the authors should include a pseudo-code in the main text.**
>
> The notation for the two equations are for the most part defined either in the preceding paragraphs or in the paragraph immediately following its respective equation. We say 'most part' because thanks to your comment we have indeed noticed two violations (noted below) of the rule that notation need be defined before appearing in equations, an issue which, while embarrassing, can be easily remedied; we apologise for our remissness in this regard and hope it this did not cause too much confusion in general.
>
> The arguments of the $NN$ function in Eq. 2 (line 179) are:
>
> - $f_\theta$: the encoder/backbone network, as defined on line 163
> - $z$:  the output of the above encoder network, as defined on line 176
> - $k$: the number of nearest-neighbours, as defined on line 177
> - $\mathcal{D}$: The aggregate dataset (union of the unlabelled and labelled training data), as defined on line 197 (**to be fixed**)
>
> Likewise, for Eq. 4 (line 206) we have:
> - $\mathbf{z}$: the set of all encodings, as defined on line 204
> - $\mathbf{e}$: the set of all associated (with $\mathbf{z}$) propensity scores, as defined on line 205
> - $\mathbf{s}$: the set of domain labels, as defined on line 229 (**to be fixed**).
>
> On the subject of pseudo code, we are happy to provide this for the matching algorithm, as well as that of the online algorithm (space-permitting), in the main text. We also intend to promote the figure illustrating the stages of the matching algorithm from Appendix D to the main text to facilitate understanding of our method, which we realise can be challenging to follow due to the number of moving parts.
>
> **In the algorithm, you tried to push the representations of the query example to be close to its unlabelled neighbors. Why would it help generalization? Aren't their representations already very close (since they are selected by k-NN)?**
>
> While k-NN has been recently shown to be an effective means of view-generation in self-supervised learning (e.g. [KTP21, DAT+21]), the crucial thing to note as far as our method is concerned is that the matching is conducted in a 'cross-domain' fashion. Ignoring the complexities of the CaliperNN algorithm, the nearest neighbours for a given query are not searched for in the dataset unconditionally, but subject to the constraint that those neighbours derive from different domains to said query -- this is formalised in Eq. 2. The intuition is that, lacking target labels, we want to bring samples that are similar, but belonging to different domains, close together in representation space, resulting in an encoder (and thus downstream predictor) that is insensitive to domain-specific features.
>
> ## References
> - [DAT+21] Debidatta Dwibedi, Yusuf Aytar, Jonathan Tompson, Pierre Sermanet, and Andrew Zisserman. With a little help from my friends:Nearest-neighbor contrastive learning of visual representations. In Proceedings of the IEEE/CVF International Conference on Computer Vision, pages 9588–9597, 2021.
> - [KTP21] Soroush Abbasi Koohpayegani, Ajinkya Tejankar, and Hamed Pirsiavash. Mean shift for self-supervised learning. In Proceedings of the IEEE/CVF International Conference on Computer Vision, pages 10326–10335, 2021.

---

> > ### Comment · Reviewer_SyKB · 2022-08-07
> > **Reply to the rebuttal**
> >
> > Thank you for addressing my questions. I also have read the updated version and other reviewers' comments. I think the empirical benefits are still far from convincing.
> > + In PovertyMap, Okapi and ERM have the same OOD performance.
> > + In iWildCam, Okapi does improve over ERM). However, is 33.3->36.1 really meaningful?
> >
> > I think the quality of this submission is currently below the acceptance rate of a NeurIPS paper. I, therefore, decide to maintain the original score for the paper.

---

> > > ### Author Response · Authors · 2022-08-08
> > > **Thank you**
> > >
> > > Thank you for taking the time and effort to read through our responses and the other reviewers comments; we're happy to hear that we were able to properly address your questions, even if there are some outstanding concerns regarding the empirical aspects of our submission.
> > >
> > > We note that previous work failed to show any significant (if any at all) improvement with respect to the ERM baseline; the WILDS 2.0 benchmark is a recent benchmark but one in which many people have already participated (https://wilds.stanford.edu/leaderboard/) without proof of consistent improvement over ERM (or even the ability to consistently match it). Indeed benchmark datasets in general have been instrumental in furthering our field: even small victories (performance increases) are victories which can serve as the foundation for future research, as epitomised by ImageNet classification where improvements of a few percent are meritable due to the extent of the dataset's study.

---

### Official Review · Reviewer_xMLR · 2022-07-08

**Rating:** 6
**Confidence:** 4
**Soundness:** 2 fair
**Presentation:** 3 good
**Contribution:** 2 fair

**Summary:**

The paper uses a modified nearest-neighbors method to match unlabeled data with the labeled data in the latent feature space for semi-supervised learning. The authors conduct experiments on iWildCam and PovertyMap datasets. They compare the performance of their proposed method to two current semi-supervised learning methods (i.e., FixMatch and ERM).

**Questions:**

Did you record the memory, computation costs, and carbon footprints of your experiments? If so, will you be able to add the analysis to the paper for the camera-ready version?

**Limitations:**

The computational and memory costs in addition to the carbon footprint of the proposed method is not discussed. As the proposed method, theoretically, seems to be computationally intensive, analysis of these costs become crucial for the assessment of the method.

**Strengths And Weaknesses:**

# Originality
The general idea of using some similarity measure to pair labeled and unlabeled images for semi-supervised learning is not new (e.g., Chen et al. (2022)). In this paper, the authors use nearest-neighbors as a measure of similarity.
# Quality
## Strengths
The paper is well written. Apart from some grammatical errors, the quality of writing is good.

## Points of concern
The proposed nearest-neighbors matching algorithm uses the feature vectors stored in a memory bank for all the previous unlabeled input images. I have concerns regarding the memory requirement for such a memory bank. Also, given the sheer size of modern unlabeled datasets (over 100k samples in the datasets studied here), using KNN to find a match for each input image among all the previous unlabeled images during training seems like a very computationally expensive approach. To address these concerns, I suggest that the authors report the memory and computation requirements for their method. Additionally, I suggest that they compare their method to a supervised learning approach not just in terms of OOD performance, but also in terms of memory and computation costs. Then plot a trade-off curve between the added costs and gain in performance for their proposed method compared to the supervised method. Then we can see whether Okapi has practical significance.

The concern about the practical significance of the proposed method is further exacerbated by the marginal improvements in the benchmarked datasets (0.9 improvement in macro F1 score for the iWildCam and 0.05 improvement in Pearson correlation for the PovertyMap). Again, extensive analysis of the added memory and computational costs of the proposed method is required to see whether it is justified to burden such additional costs in exchange for the minor improvements in performance.


# Clarity
The overall clarity of the paper is good. There are some grammatical errors that need to be fixed (e.g., line 333: “each which…”)
# Significance
Proposing new algorithms without proper cost analysis proves little practical significance.

---

> ### Author Response · Authors · 2022-08-02
> **Response to review**
>
> **The general idea of using some similarity measure to pair labeled and unlabeled images for semi-supervised learning is not new (e.g., Chen et al. (2022)).**
>
> While it is unclear to us which paper the reviewer is referring to in their example (we would greatly appreciate the full citation so we might investigate the extent of the similarities), we are fully aware that we are by no means to leverage the idea of matching labelled and unlabelled data for semi-supervised learning -- the novelty of our method is not in the general premise but in the idea of using statistical matching to perform cross-domain matching. It is worth mentioning that the matching is not constrained to be between labelled and unlabelled data -- though that is the mode we explore in the paper -- but between domains. As such, our method is applicable even when the domains of the two aforementioned subsets have intersecting support, providing that information about the domain from which a given sample belongs to is made available (which is far more likely in practice than it is for the targets).
>
> **I have concerns regarding the memory requirement for such a memory bank. Also, given the sheer size of modern unlabeled datasets (over 100k samples in the datasets studied here), using KNN to find a match for each input image among all the previous unlabeled images during training seems like a very computationally expensive approach. To address these concerns, \red{I suggest that the authors report the memory and computation requirements for their method. Additionally, I suggest that they compare their method to a supervised learning approach not just in terms of OOD performance, but also in terms of memory and computation costs. Then plot a trade-off curve between the added costs and gain in performance for their proposed method compared to the supervised method. Then we can see whether Okapi has practical significance.**
>
> There seems to be some misconception regarding the workings of the memory banks (something we will try to avoid further instances of by improving the clarity in this regard, both illustratively and textually). The memory banks do not store the encodings and labels of all previously visited samples -- something which would indeed worrisome due to implying linear time and memory complexities of CaliperNN w.r.t. the total number of samples $N := N_l + N_u$ -- rather these are fixed-size FIFO queues and as such store only the last $N_M$ instances of such pairs. In the case of iWildCam and PovertyMap, we set $N_M$ to $2^{14}$ and $2^{16}$, respectively, corresponding to $\sim 1.6\%$ and $\sim 33\%$ of $N$ for their respective datasets. Indeed, it is precisely the intractability of performing a nearest-neighbour search over modern datasets which we seek to obviate with our online procedure -- after the memory banks have been populated (i.e. following visitation of $N_M$ samples), the memory and computational requirements of the algorithm remain constant for the remainder of training. Our method does introduce additional overhead compared with other memory-bank-based approaches, such as MoCo [HFW+20] in that a memory bank is used not only to store the encodings, but also the domain labels. This overhead, however, is comparatively small given that the labels can be cheaply stored in the form of one-dimensional integer tensors.
>
> The aforementioned aside, we do think the suggestion of reporting the per-iteration cost (in terms of time and memory), along with the per-run carbon footprint, of all methods (Okapi and baselines alike) is a salient one, one that will helpfully bolster confidence in our proposed method which has the advantage of being computationally more efficient than such self-training methods such as FixMatch and Noisy Student (the latter in particular due to the multiple rounds of training involved).

---

> > ### Comment · Reviewer_xMLR · 2022-08-08
> > **comment on the rebuttal**
> >
> >
> > [WNG21] propose Neighbor Matching, in which they find neighbouring samples for an unlabelled instance. Similar to your work, I believe, Neighbor Matching also uses the embedding of the samples to find neighbours.
> >
> > Regarding the experimental results, it appears that other reviewers also have concerns about the marginal or lack of improvement of Okapi over previous methods. At this point, Okapi seems like a theoretically interesting work that lacks practical significance.
> >
> > On your comments about computation costs, I appreciate your acknowledgment of its importance. Unfortunately, I did not see any action or motivation to address that.
> >
> > ### References
> > - [WNG21] Wang, R., Wu, Y., Chen, H., Wang, L., & Meng, D. (2021, September). Neighbor Matching for Semi-supervised Learning. In International Conference on Medical Image Computing and Computer-Assisted Intervention (pp. 439-449). Springer, Cham.

---

> > > ### Author Response · Authors · 2022-08-09
> > > **Thank you**
> > >
> > > Thank you for taking the time to read our responses and those of the other reviewers; we greatly appreciate your feedback and your providing the reference (despite it being behind a paywall). Said reference (Wang et al.) is indeed similar in the respects that it leverages k-NN and a memory bank for semi-supervised learning, however there are some key distinctions we would like to point out: 1) Okapi uses an augmented form of k-NN in CaliperNN which helps improve robustness to outliers; when the fixed and std calipers are set to 1 and $\infty$ respectively, we recover regular NN; 2)  In Okapi the consistency loss is imposed on the encodings rather than on the prediction meaning that it is generalisable to any modality, whereas the variant proposed by Wang et al. is strictly suited for classification (with k-NN + attention module being used to generate pseudo-labels for a second cross-entropy loss); 3) In Okapi we tackle the problem of semi-supervised domain generalisation, rather than semi-supervision in general, which leads us to imposing a cross-domain constraint on the matching; 4) In addtition to being classification-only, the datasets  explored in Wang et al. are limited to medical data domain, whereas we are able to demonstrate success on a range of datasets and tasks featuring different kinds of natural distribution shifts; 5) Okapi uses a slow-moving momentum encoder to generate the keys in the memory bank — this helps stabilise matching by preventing said keys from changing too quickly and the query-key distances thereby losing meaning.. This all said, we would be willing to include results with the neighbour-matching method of Wang et al. at the reviewer’s insistence, though with the caveat that it is not applicable to PovertyMap due its being a regression task.
> > >
> > > Regarding the carbon footprint of our experiments, we have taken steps to address your concerns by estimating the amount of carbon emitted by a single controlled training run of each method on the iWildCam dataset (tabulated below); the same statistics will be computed for each dataset for the revised version of the paper. For this training,  we used a private infrastructure with an estimated carbon efficiencyof 0.432 kgCO$_2$eq/kWh and RTX 3090 GPUs. Each job was run on a single GPU and with four data-loading workers.
> > >
> > > | Method|kgCO$_2$eq |
> > > |-|-|
> > > | ERM | 1.36 |
> > > | Okapi | 1.97 |
> > > | FixMatch | 2.12|

---

### Official Review · Reviewer_mbZt · 2022-07-10

**Rating:** 6
**Confidence:** 3
**Soundness:** 2 fair
**Presentation:** 3 good
**Contribution:** 3 good

**Summary:**

The paper proposes a general and robust semi-supervised learning algorithm called Okapi that works for data from various domains. It is based on the simple idea of supplementing the supervised loss with a cross-domain consistency loss that encourages the outputs of an encoder network to be similar for k-nearest neighbor samples belonging to different domains. To make the online matching in a runtime- and memory-efficient way, they incorporates a statistical matching framework and fixed-size memory bank. The results show that Okapi outperforms the baseline methods in terms of out-of-distribution (OOD) generalization on iWildCam and PovertyMap datasets. The work suggests that it is possible to effectively incorporate unlabelled data and domain information into a training algorithm to improve OOD generalization.

**Questions:**

1. What is the intuition behind the CaliperNN algorithm to mimic the standard nearest neighbor algorithm?

The first stage is to remove the samples with a propensity score exceeding a predefined threshold. If I understand correctly, that means we want to remove those samples with high predictive confidence in predicting domains. In other words, we want to keep those samples that lie on the domain decision boundary. How could those samples be more likely the nearest neighbors?

The second stage of filtering is to keep paired samples from different domains only if their Euclidean distance between their respective propensity scores is below a threshold. Do you assume that the small distance in propensity score will reflect small distance in their representations?

2. There are many moving parts and hyperparameters in the algorithm. Appendix C studies the effect of choosing the number of nearest neighbours k, the use of the fixed- and std-calipers in CaliperNN, and which encoder (online or target) is used to generate the queries for statistical matching. How did you choose the threshold t_f and t_theta, and are the results sensitive to the threshold values? Moreover, since the ablation is done in PovertyMap dataset, is the conclusion transfer to other datasets like iWildCam?



**Limitations:**

yes

**Strengths And Weaknesses:**

Originality: Making use of the domain information and using propensity score to predict domain for efficient matching is novel.

Quality: The experiments are done for comparison with the existing methods, and for ablation study are solidly done.

Clarity: The paper is well written.

Significance: Most of the results in Table 1 are statistically significant.

---

> ### Author Response · Authors · 2022-08-02
> **Response to review**
>
> **What is the intuition behind the CaliperNN algorithm to mimic the standard nearest neighbor algorithm?**
>
> The CaliperNN algorithm does not attempt to mimic the standard nearest-neighbour algorithm (k-NN), but rather augments it using principles from statistical matching in order to filter out 'unlikely' matches, according to the learned propensity scores. The first stage of the algorithm is to compute distance between the all query samples and all key samples (stored in the memory bank) subject to the cross-domain constraint. The second stage is then to use the propensity scores to eliminate samples in which the propensity scorer is overly confident (determined by the fixed threshold) and eliminate matches for which its scores differ (according to Euclidean distance) by more than an adaptive threshold (based on the number of standard deviations).
>
> **The first stage is to remove the samples with a propensity score exceeding a predefined threshold. If I understand correctly, that means we want to remove those samples with high predictive confidence in predicting domains. In other words, we want to keep those samples that lie on the domain decision boundary. How could those samples be more likely the nearest neighbors?**
>
> Matching is performed strictly across domains ('cross-domain'): if we only keep the samples around the decision boundary area it will naturally result in matched pairs that are closer in the propensity-score space.
>
> **The second stage of filtering is to keep paired samples from different domains only if their Euclidean distance between their respective propensity scores is below a threshold. Do you assume that the small distance in propensity score will reflect small distance in their representations?**
>
> There is no guarantee that a small distance in propensity score will reflect small distance in their representations -- indeed, it is not the purpose of the score is not to serve as a proxy for the encodings. There are two main (good) reasons for using a propensity score as the basis for filtering rather than simply the encodings: 1) it summarises all covariates (encoded dimensions) into one value [Stu10] and is therefore a simpler aggregate measure; 2) it is a balancing score [RR83], which means that at each value of the score, the distribution of the covariates will be similar in the treatment and control groups.
>
>
> **There are many moving parts and hyperparameters in the algorithm. Appendix C studies the effect of choosing the number of nearest neighbours k, the use of the fixed- and std-calipers in CaliperNN, and which encoder (online or target) is used to generate the queries for statistical matching. How did you choose the threshold $t_f$ and $t_{\theta}$, and are the results sensitive to the threshold values? Moreover, since the ablation is done in PovertyMap dataset, is the conclusion transfer to other datasets like iWildCam?**
>
> We describe our procedure for selecting the set of hyperparameters, $\xi$ (that includes $t_f$ and $t_{\sigma}$) for CaliperNN in Appendix A, 'Implementation details', but restate it below for the reviewer's convenience.
>
> In the interest of computational efficiency, we refrain from conducting an inner-loop of optimisation and perform a grid-search over various configurations of $\xi$ in the offline setting, using a (frozen) pretrained (via ERM) encoder to generate the queries and keys for matching. To generate the search space, we trial values from the set $\{ 0.7, 0.85,  0.9, 0.95, 1.0 \}$ for the fixed caliper, $t_f$, values from the set $\{0.2, 0.4, 0.6, \infty \}$ for the std. caliper, $t_\sigma$, and values from the set $\{0.7, 1.0, 1.2, 1.5 \}$ for the temperature parameter, $\tau$. The quality of matching for a given configuration is then measured according two standard metrics in the statistical matching literature: _Variance Ratio_ (VR) and _Standard Mean Differences_ (SMD) [Rub01]. For a given pair of domains, VR is defined as the ratio of the variances of the covariates between the two domains, with an ideal value of $1$, while SMD is defined as the difference in their covariate means, normalised by the standard deviation for each covariate, and is to be minimised.
>
> ## References
> - [Stu10] Elizabeth A. Stuart. Matching methods for causal inference: A review and a look forward. Stat Sci, 25(1):1–21, 2010.
> - [RR83] Paul R Rosenbaum and Donald B Rubin. The central role of the propensity score in observational studies for causal effects. Biometrika, 70(1):41–55, 1983.

---

### Official Review · Reviewer_88f2 · 2022-07-16

**Rating:** 6
**Confidence:** 3
**Soundness:** 3 good
**Presentation:** 2 fair
**Contribution:** 2 fair

**Summary:**

The paper proposes Okapi, a method for domain adaptation / domain generalization based on the idea of matching unlabeled examples to labeled in-distribution examples and enforcing consistency in predictions. The authors propose an elaborate strategy for matching examples, and show promising performance on two datasets from the Wilds benchmark.

**Questions:**

Question 1. Please see my questions in the previous section.

Question 2. Is there a reason to train the propensity classifier online, along with the encoder / main classifier? It appears that you can train it once before training the main classifier, and compute all matchings, avoiding the complications with the memory bank and online training? Or is the fact that the propensity classifier using the same features as the main classifier important to the method?

**Limitations:**

No issues.

**Strengths And Weaknesses:**

Strength 1. The paper tackles a challenging problem of using unlabeled data for improving out-of-distribution performance. Prior work suggested that it is not trivial to achieve performance improvements over standard ERM by using unlabeled data. The authors show promising results.

Strength 2. On a high level, the method makes intuitive sense. The authors also perform an ablation evaluating the importance of some parts of the method.

Weakness 1. The method is quite complex, with a lot of moving parts. It involves two versions of an encoder (regular and with smoothed weights), propensity score classifier based on the embeddings, regular classifier, memory banks, two different criteria for discarding datapoints from matching (referred to as calipers in the paper). The method is not easy to understand, and it is not presented very clearly. The two calipers (methods for discarding outlier datapoints), std and fixed, are not described in sufficient detail in the main text. The matching strategy is also only described somewhat vaguely. In particular, what is $\xi$ in CaliperNN$_{\xi}$? How exactly are propensity scores used?

Given the complexity of the approach, it would be useful to have an algorithm box explaining precisely what each stage of the procedure is doing.

Weakness 2. The empirical evaluation is somewhat limited. The authors focus on only two (challenging) datasets from the Wilds benchmark, comparing to two baselines: ERM and FixMatch. FixMatch was shown by prior work to perform poorly on these datasets, as mentioned by the authors. It is hard to know how significant the improvement is relative to the baselines: on poverty map ERM performs comparably to Okapi in terms of MSE. On iWildCam FixMatch (which was reported to work quite poorly on this dataset according to the authors) performs comparably to Okapi.

On both datasets, Okapi performs slightly better, but it is hard to tell if the improvement is significant, especially given that the method is more or less designed to work well on these two datasets, while FixMatch was shown to perform particularly poorly on iWildsCam.

Another complication with the evaluation is that the authors chose to use a different backbone compared to prior work, ConvNext. Consequently, we cannot directly compare the numbers reported by the authors to the numbers reported in the prior work. Why was this choice made?

The Calipers in table 1 appear to provide a fairly minor improvement compared to Okapi no caliper.

Weakness 3. It seems like the method may not work well in the cases when the domains are sufficiently different, such as in the common domain generalization benchmarks. The idea of the method is to reject datapoints which can be differentiated from the in-distribution datapoints (based on propensity scores). On datasets such as PACS the domains can be clearly differentiated, so my understanding is that the method would reject all the images from other domains. Is this interpretation correct?

---

> ### Author Response · Authors · 2022-08-02
> **Response to review (1)**
>
> **The method is quite complex, with a lot of moving parts. It involves two versions of an encoder (regular and with smoothed weights), propensity score classifier based on the embeddings, regular classifier, memory banks, two different criteria for discarding datapoints from matching (referred to as calipers in the paper).**
>
> Implementation-wise, our method largely builds upon the momentum-encoding paradigm, as popularised by MoCo [HFW+20], only with the addition of another memory bank for storing the domain-labels (which are simply integers and contribute comparatively little memory overhead) alongside the encodings; the bulk of the complexity comes from the CaliperNN algorithm, for which we intend to provide detailed pseudocode for; the implementation itself will be made publicly available alongside the rest of the code/experiment-running scripts upon publication. Last but not least, our experiments with CivilComments dataset where we  use the same hyper-parameter setup as in the main paper, show that Okapi is robust to the choice of hyperparameters and can be used in an off-the-shelf manner.
>
> **The method is not easy to understand, and it is not presented very clearly. The two calipers (methods for discarding outlier data-points), std and fixed, are not described in sufficient detail in the main text.  The matching strategy is also only described somewhat vaguely. Given the complexity of the approach, it would be useful to have an algorithm box explaining precisely what each stage of the procedure is doing.**
>
> Our arguably wrong decision to defer it to the supplementary was due to the spatial constraints. We are happy to incorporate the suggestion of the reviewer to include this information in the main text, e.g. as an algorithm box or a truncated version of text. Apropos of the reviewer's concern about the calipers, we agree with them and will be sure to include a more detailed explanation of these elements of the algorithm in the updated version of the text.
>
> **The empirical evaluation is somewhat limited}.The authors focus on only two (challenging) datasets from the Wilds benchmark, comparing to two baselines: ERM and FixMatch. FixMatch was shown by prior work to perform poorly on these datasets, as mentioned by the authors. It is hard to know how significant the improvement is relative to the baselines: on poverty map ERM performs comparably to Okapi in terms of MSE. On iWildCam FixMatch (which was reported to work quite poorly on this dataset according to the authors) performs comparably to Okapi.**
>
> To address the current limitations in our empirical evaluation we will be adding results for at least one more dataset in Civil Comments, preliminary results for which we have included in our 'response to all reviewers'.
>
> **On both datasets, Okapi performs slightly better, but it is hard to tell if the improvement is significant, especially given that the method is more or less designed to work well on these two datasets, while FixMatch was shown to perform particularly poorly on iWildsCam**
>
> Based on the benchmark report by [SKL+22], ERM was found to vastly outperform any competing methods utilising the unlabelled data and/or domain labels on the PovertyMap dataset; Okapi performs slightly better than ERM in terms of corr. metric (0.55 versus 0.5), while maintaining the performance in terms of MSE when using unlabeled data as opposed to FixMatch, 0.33 MSE versus 0.88. Nevertheless, our Okapi outperforms both methods, with 36.1 F1 score (OOD). We also note that we significantly upgraded the FixMatch baseline for iWildCam using the ConvNeXT backbone, achieving 35.2 F1 score (OOD), versus the 31.0 of [SKL+22] . This shows the importance of using a newer backbone for domain generalisation (see the next response for more discussion regarding the backbone). Finally, we dispute the claim that Okapi was designed to work for iWildCam and PovertyMap specifically: the method is a general one, both in terms of modality and the range of domain generalisation setups to which it can be applied. We point the reviewer to our Civil Comments results as evidence of this.

---

> ### Author Response · Authors · 2022-08-02
> **Response to review (2)**
>
> **Another complication with the evaluation is that the authors chose to use a different backbone compared to prior work, ConvNext. Consequently, we cannot directly compare the numbers reported by the authors to the numbers reported in the prior work. Why was this choice made?**
>
> This choice was motivated by the preliminary observation that while FixMatch performs worse than ERM in with the ResNet backbone employed by [SKL+22], this does not hold when switching to a ConvNext backbone, as evidenced in our results with FixMatch now outperforming the ERM baseline. Such an observation is in line with the concurrent work of [KWSS22] who similarly noticed the choice of backbone to be a significant determining factor in domain generalisation. We speculate that Batch Normalisation (BN) is at least partly to blame for this discrepancy due to the problems that emerge from applying statistics accumulated from the seen domains (training data) to unseen domains. Indeed, a number of previous works have identified the vulnerability of BN to distribution shifts and have proposed variants, such as TransNorm [WJL+19] and MetaNorm [DZSS21] in an attempt to redress it. Unlike ResNet, ConvNext, adopts layer normalisation as the normalisation method of choice throughout and through it we sidestep the aforementioned issue.
>
> **The Calipers in table 1 appear to provide a fairly minor improvement compared to Okapi no caliper.**
>
> Although the improvement is minor in term of OOD correlation (+0.01), we believe that one can still benefit from the usage of caliper by looking at the results in aggregate: a slightly better OOD correlation, better OOD MSE (-0.03), together with a reduced standard deviation across runs -- these all point to the calipers aiding robustness.
>
> **What is in CaliperNN? How exactly are propensity scores used?}**
>
> CaliperNN can be summarised as follows: 1) we estimate the propensity score and re-scale it with a fixed temperature 2) we restrict potential matches based on fixed caliper applied to the propensity score and 3) we select the closest sample using Euclidean distance matching while defining a maximum propensity score distance between samples. We call it CaliperNN due to it being an hybrid approach that uses two calipers applied to the estimated propensity score distribution (fixed caliper in stage 2 and std. caliper in stage 3) in order to determine suitable matches, as in statistical matching, along with nearest-neighbour based retrieval (in feature space), as in NN.
>
> **The idea of the method is to reject datapoints which can be differentiated from the in-distribution datapoints (based on propensity scores). On datasets such as PACS the domains can be clearly differentiated, so my understanding is that the method would reject all the images from other domains. Is this interpretation correct?**
>
> The user can change the fixed caliper threshold and temperature scaling to smooth the propensity score distribution to prevent the algorithms from discarding all samples. We performed some initial exploratory experiments on PACS dataset (using features extracted for a pre-trained CLIP [RKH+21] visual encoder) to show how this holds true in the cases highlighted by the reviewer. We tabulate below the number of matched pairs retrieved when matching across the two domains _photo_ and _sketch_; here we can see that by increasing the temperature we smooth the estimated propensity score distribution and thereby retain more samples. Similarly, we can retrieve more pairs by reducing the fixed caliper threshold.
>
> | Fixed Caliper ($t_f$) | Temperature ($\tau$) | photo $\rightarrow$ sketch | sketch $\rightarrow$ photo|
> | ----------- | ----------- | ----------- | ----------- |
> | 0.9 | 1       | 0    | 0    |
> | 1.0 | 1       | 1540 | 3929 |
> | 0.99 | 1       | 6    | 9    |
> | 0.99 | 1.3     | 14   | 56   |
> | 0.99 | 1.8     | 25   | 574  |
> | 0.99 | 2.5     | 41   | 3082 |
> | 0.99 | 10      | 1540 | 3929 |
> | 0.9 | 10      | 298  |  3929|
>
> We also analyse the case of matching across the two domains _photo_ and _art painting_. Using a fixed caliper defined defined by a threshold $t_f=0.9$ and no temperature scaling (i.e. $\tau=1$) the algorithm retrieves 1,142 pairs matching in the direction photo $\rightarrow$ art and 1,501 in the direction art $\rightarrow$ photo. A figure showing the shape of the estimated propensity score as well as examples of matched pairs can be found in Appendix E. Although the encodings used were not the result of fine-tuning on the PACS dataset, we are still able to recover a few intraclass matches. For the photo-art painting application we can see preservation in colours and background; in the photo-sketch case, shape and pose.

---

> ### Author Response · Authors · 2022-08-02
> **Response to review (3)**
>
> **Is there a reason to train the propensity classifier online, along with the encoder/main classifier It appears that you can train it once before training the main classifier, and compute all matchings, avoiding the complications with the memory bank and online training? Or is the fact that the propensity classifier using the same features as the main classifier important to the method?**
>
> One could envision an offline version of the algorithm in which one simply computes the matches using a pre-trained encoder and then trains a classifier by enforcing consistency between those pre-defined matches (fixed over the course of training). In fact, our initial experimentation adopted precisely this version of the algorithm, however we found that bootstrapping the algorithm by computing the matches in an online fashion yielded significantly better results while only requiring matches to be computed between a small subset of the data at any given time, thanks to the combination of momentum-encoding and the memory bank, and while streamlining the training process into a single stage. Indeed, such use of bootstrapping by using clustering/matching to generate a proxy task has found great success in the self-supervised learning literature, as we discuss in our related work section. That said, we do believe that the aforementioned offline algorithm would serve as a useful baseline/ablation, in retrospect, and would be sure to be entered into the camera-ready version of the paper. Even with the matching done online, it is still possible to train the propensity scorer, in an inner loop, to convergence on the current set of past and present encodings, however we found this to be prohibitively expensive in practice, especially when employing larger-size memory banks.
>
> ## References
> - [DZSS21] Yingjun Du, Xiantong Zhen, Ling Shao, and Cees G. M. Snoek. Metanorm: Learning to normalize few-shot batches across domains. In International Conference on Learning Representations, 2021.
> - [HFW+20] Kaiming He, Haoqi Fan, Yuxin Wu, Saining Xie, and Ross Girshick. Momentum contrast for unsupervised visual representation learning. In Proceedings of the IEEE/CVF conference on computer vision and pattern recognition, pages 9729–9738, 2020.10
> - [KWSS22] Donghyun Kim, Kaihong Wang, Stan Sclaroff, and Kate Saenko. A broad study of pre-training for domain generalization and adaptation. arXiv preprint arXiv:2203.11819, 2022.
> - [RKH+21] Alec Radford, Jong Wook Kim, Chris Hallacy, Aditya Ramesh, Gabriel Goh, Sandhini Agarwal, Girish Sastry, Amanda Askell, Pamela Mishkin, Jack Clark, et al. Learning transferable visual models from natural language supervision. In International Conference on Machine Learning, pages 8748–8763. PMLR, 2021.
> - [SKL+22] Shiori Sagawa, Pang Wei Koh, Tony Lee, Irena Gao, Sang Michael Xie, Kendrick Shen, Ananya Kumar, Weihua Hu, Michihiro Yasunaga, Henrik Marklund, Sara Beery, Etienne David, Ian Stavness, Wei Guo, Jure Leskovec, Kate Saenko, Tatsunori Hashimoto, Sergey Levine, Chelsea Finn, and Percy Liang. Extending the WILDS benchmark for unsupervised adaptation. In International Conference on Learning Representations, 2022.
> - [WJL+19] Ximei Wang, Ying Jin, Mingsheng Long, Jianmin Wang, and Michael I Jordan. Transferable normalization: Towards improving transferability of deep neural networks. Advances in neural information processing systems, 32, 2019

---

> > ### Comment · Reviewer_88f2 · 2022-08-08
> > **Thank you for the rebuttal**
> >
> > I appreciate all the additional effort you put into the clarifications and new experimental results!
> >
> > The results on the CivilComments are encouraging, and improve upon the best results for ERM reported on the WILDS leaderboard.
> >
> > For the results on the other benchmarks, it is somewhat challenging to compare the results reported to those on the Wilds leaderboard due to the change of backbone. I think the results are encouraging, however. Ideally, I would like to see results using the same backbone as the prior work (in addition to the current results) to make sure that the ERM baseline was not undertuned for the ConvNext backbone.
> >
> > Thank you for including preliminary results on the PACs dataset!
> >
> > Thank you for the comments clarifying the methodology. I think including an algorithm box will strengthen the paper and make it easier to follow.
> >
> > Given the comments above, I increase my score to a weak accept (6).

---

> > > ### Author Response · Authors · 2022-08-08
> > > **Thank you!**
> > >
> > > Thank you very much, both for taking the time to read our responses and the updated supplementary material, and for the increase in score in light of it  -- we're glad that we were able to address your concerns, particularly as relates to the clarity of the CaliperNN algorithm. We agree that adding an algorithm box would be a compact way to aid understanding of what is an admittedly-quite-complicated method and will be sure to do so for the revised version of the paper. We also agree that results for an ERM baseline using the same backbone architecture (ResNet-50) as prior work would lend empirical justification to our decision to use an alternative architecture and will be certain to include them along with any other additional results.

---

### Author Response · Authors · 2022-08-02
**Response to all reviewers**

We would begin by thanking all reviewers for taking the time to attentively read, and leave feedback -- both positive and negative alike -- on, our submission. We agree that there is room for improvement with respect to the clarity, contextualisation, and empirical justification of our method but believe that these things require only minor revisions and that, on the whole, our paper is well-written and easy-to-understand (R. xMLR, R. mbZt) and proposes a method that is novel (R. mbZt), interesting (R. SyKB), intuitive (R. 88f2) and with practical merit (R. 88f2).

# Civil Comments
There was shared concern about the extensiveness of the empirical evaluation; to help allay this we provide below preliminary results on the WILDS 2.0 version of the Civil Comments dataset [SKL+22], both for our method (Okapi) with the ERM baseline. We kept the architecture (BERT uncased) and data-loading settings consistent with those in [SKL+22] across the board and for the latter method (ERM) used the optimisation settings reported in the CodaLab worksheets (https://worksheets.codalab.org/worksheets/0xff0ec35397fc44319f9a4ef8071056ea) given for reproducing the experiments in [SKL+22], with the base configuration derived from the official WILDS repository(https://github.com/p-lambda/wilds/tree/main/examples). We hope that these results help convince reviewers as to the robustness and generality of our approach, particularly in its support of our claims of the modality-agnostic nature of Okapi.

| Method | worst-group acc (OOD) |
| - | - |
| ERM [SKL+22] | 66.6 (1.6)|
| ERM (fully-labeled) [SKL+22] | 69.4 (0.6)|
| ERM (reproduction)  | 68.5 (2.2) |
| Okapi (ours)| 69.7 (2.0) |

As the official splits of the dataset do not comprise of an ID version of the test split, and we were unable to find details of how said split was generated for the results in [SKL+22] (neither in the paper nor in the aforementioned codebase), we report here only performance on the OOD test set to ensure comparability. Despite using identical hyperparameters to [SKL+22] for ERM, we observe  marked improvement in the worst-group accuracy of this baseline compared with that reported therein. We attribute this partly to the decision to validate every 1k iterations -- amounting to a roughly five-fold increase in frequency compared with validating after every epoch, as done in [SKL+22]; this is of consequence to the test results due to their being produced by the best-performing checkpoint based on the validation split (we follow [SKL+22] in this regard). This aside, we observe that with minimal changes in configuration Okapi outperforms the ERM baseline in terms of both expectation and variance by a respectable margin, to the point of being on par with the 'fully-labelled' baseline from [SKL+22]

# References
- [SKL+22] Shiori Sagawa, Pang Wei Koh, Tony Lee, Irena Gao, Sang Michael Xie, Kendrick Shen, Ananya Kumar, Weihua Hu, Michihiro Yasunaga, Henrik Marklund, Sara Beery, Etienne David, Ian Stavness, Wei Guo, Jure Leskovec, Kate Saenko, Tatsunori Hashimoto, Sergey Levine, Chelsea Finn, and Percy Liang. Extending the WILDS benchmark for unsupervised adaptation. In International Conference on Learning Representations, 2022.

---

### Meta-Review · Program_Chairs · 2022-09-14

**Recommendation:** Accept
**Confidence:** Less certain

**Metareview:**

in this paper, the authors propose an algorithm called Okapi that improves out-of-domain generalization by ensuring the representation similarity between in-domain (labelled) examples and out-of-domain (unlabelled) examples. despite the algorithm's high complexity, which initially concerned the reviewers, the empirical results (both in the original and response) as well as the authors' successful rebuttal convinced the reviewers that this manuscript is worth publication. therefore, i recommend acceptance.

authors, please take into account the comments from the reviews when preparing your camera-ready version.

**Award:**

No

---

### Decision · Program_Chairs · 2022-09-14

Accept